# Healthy LLMs? Benchmarking LLM Knowledge of UK Government Public Health Information

## Abstract

As Large Language Models (LLMs) become widely accessible, a detailed understanding of their knowledge within specific domains becomes necessary for successful real world use. This is particularly critical in the domains of medicine and public health, where failure to retrieve relevant, accurate, and current information could significantly impact UK residents. However, while there are a number of LLM benchmarks in the medical domain, currently little is known about LLM knowledge within the field of public health. To address this issue, this paper introduces a new benchmark, PubHealthBench, with over 8000 questions for evaluating LLMs' Multiple Choice Question Answering (MCQA) and free form responses to public health queries. To create PubHealthBench we extract free text from 687 current UK government guidance documents and implement an automated pipeline for generating MCQA samples. Assessing 24 LLMs on PubHealthBench we find the latest proprietary LLMs (GPT-4.5, GPT-4.1 and o1) have a high degree of knowledge, achieving over 90% accuracy in the MCQA setup, and outperform humans with cursory search engine use. However, in the free form setup we see lower performance with no model scoring over 75%. Therefore, while there are promising signs that state of the art (SOTA) LLMs are an increasingly accurate source of public health information, additional safeguards or tools may still be needed when providing free form responses.

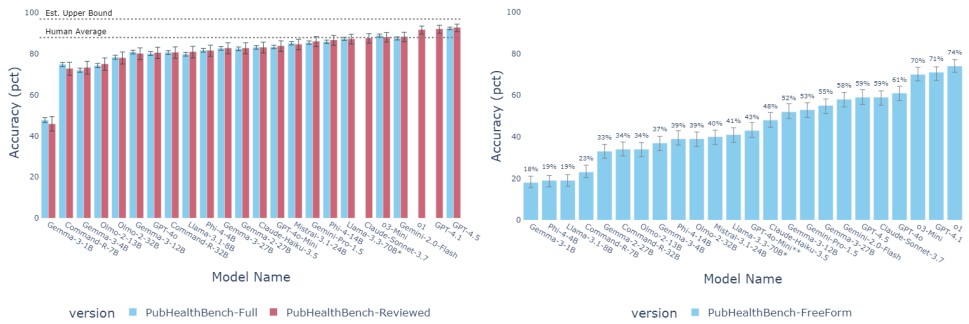

Figure 1: (left) PubHealthBench Full and Reviewed model accuracy, (right) PubHealthBench-FreeForm model accuracy. 95% Wilson CI. *LLM used to generate benchmark, **Judge LLM.

## 1 Introduction

Public health guidance represents an important source of information for UK residents and experts to inform personal, professional, and clinical decision making. The release of highly capable Large Language Models (LLMs) (Minaee et al., 2025), and particularly chatbots (OpenAI, 2022), could represent a significant shift in how public health guidance is retrieved, analysed, and disseminated. This in turn raises significant opportunities and risks for public health institutions both internally and when engaging the public.

Whilst LLMs often undergo a broad range of evaluations during development (Grattafiori et al., 2024; Google, 2025; OpenAI, 2025b; Anthropic, 2025), and there are a number of existing benchmarks in the medical domain, there is currently an important gap in field of public health, with no comprehensive LLM benchmarks covering this domain, including for existing UK Government guidance. Furthermore, due to guidance undergoing regular revisions, and differing guidance being issued across institutions and geographies, accurate up to date knowledge of UK public health guidance may be particularly challenging for LLM systems. Therefore, as recently observed for BBC news stories (BBC, 2024), there is a risk that LLM based applications and chatbots generate hallucinations (Huang et al., 2025) or incomplete information regarding UK public health advice. This in turn could have a significant impact on the public. These risks, combined with the increasing desire within the UK Government to incorporate LLMs into existing real world processes (Department for Science & Technology, 2023; UKHSA, 2023), means comprehensive evaluations of LLMs' understanding of UK public health guidance are needed.

In this paper we introduce a new dataset, Multiple-Choice Question Answering (MCQA) benchmark, and free form response benchmark for assessing LLMs' knowledge across a broad range of UK public health guidance.[1] Specifically our contributions include:

**PubHealthBench a fully grounded MCQA benchmark** - We collect, extract, markdown format, and chunk information from over 500 publicly available UK Health Security Agency (UKHSA) PDF and HTML documents from the UK Government website (gov.uk).[2] We implement an automated MCQA generation and validation pipeline grounded in the extracted guidance source text. This enables us to generate a new benchmark with over 8000 MCQA questions to test LLM knowledge across a broad range of current guidance. We provide results for both the full benchmark (PubHealthBench-Full) and a manually reviewed subset (PubHealthBench-Reviewed).

**PubHealthBench-FreeForm** - To assess LLMs in a more realistic real-world setting we also implement a free form response benchmark using the questions from the manually reviewed subset. By utilising the fact that every question can be linked back to the original source chunk and document, we implement a grounded LLM judge to assess responses.

**Initial LLM evaluations** - We evaluate 24 private and open-weight LLMs on this new public health benchmark. Given our focus on assessing knowledge, we primarily focus on SOTA non-reasoning models, but also include some leading reasoning models for comparison.

**Manual human expert review and human baseline** - To quality assure the benchmark and establish an upper bound for performance, human experts manually reviewed a random sample of 800 MCQA examples (approx. 10% of the benchmark). Furthermore, in order to compare to human performance, 5 humans also took sample tests (total 600 MCQA examples) to establish a human baseline with the use of search engines. This provides an initial indication of the boundary at which LLMs become similarly accurate to a cursory search by non-expert humans.

## 2 RELATED WORK

### 2.1 MCQA LLM EVALUATIONS

Using Multiple-Choice Question Answering (MCQA) to assess the knowledge of LLMs is well established in the literature. Earlier work evaluating LLM knowledge in specific domains often used existing MCQA human assessments and exams (OpenAI, 2023; Bommarito et al., 2023; Jin et al., 2020; Pal et al., 2022), for example in the medical domain using evaluations from the US Medical Licensing Examination (USMLE) (Kung et al., 2023; Singhal et al., 2023).

Broader evaluations of LLM knowledge and capabilities have also been introduced. One of the most widely adopted being the Massive Multitask Language Understanding (MMLU) benchmark (Hendrycks et al., 2021). More recently the MMLU-Pro benchmark by Wang et al. (2024), updated the original MMLU evaluations for errors (Gema et al., 2024) and increased the ques-

---

[1]The guidance included can cover the entire UK or individual constituent countries, most documents primarily relate to English public health guidance. We also only included English language documents.

[2]GOV.UK reuse policy: https://www.gov.uk/help/reuse-govuk-content

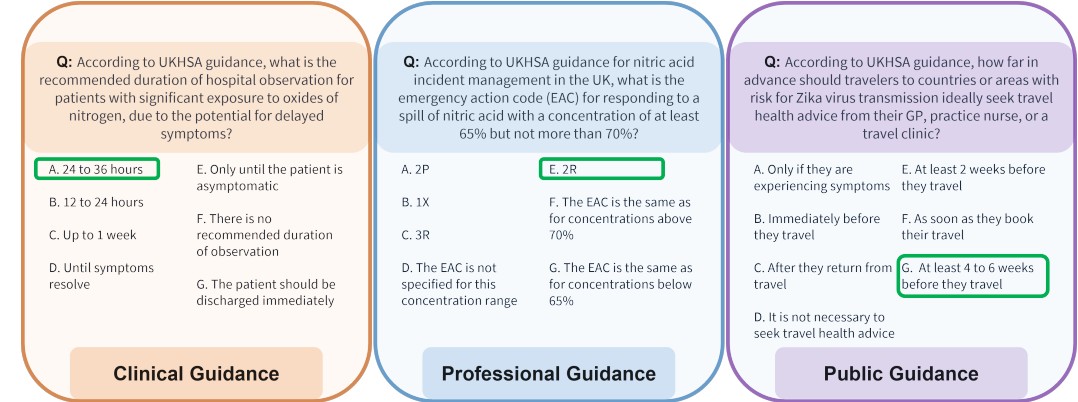

Figure 2: Example PubHealthBench MCQA benchmark questions.

tion difficulty. New domain specific MCQA LLM evaluations have also been created, such as the GPQA (Rein et al., 2023) and ARC (Clark et al., 2018) benchmarks within science.

## 2.2 SYNTHETIC MCQA DATASETS

There is increasing research investigating automated approaches to MCQA generation grounded in existing corpora (Shashidhar et al., 2025; Guinet et al., 2024; Ghazaryan et al., 2024). To develop MMLU-Pro, Wang et al. (2024) use GPT4-Turbo to generate answer options, as well as to augment the option lists found in other LLM benchmarks (primarily MMLU) with additional distractors. Similarly a combined human-LLM pipeline is used to construct the recent SuperGPQA benchmark (Team et al., 2025) and in Asiedu et al. (2025)'s work evaluating LLMs for tropical and infectious diseases. Concurrent work by Shashidhar et al. (2025) on the YourBench framework goes further and provides a fully automated approach to generating MMLU style evaluations from new document corpora.

## 2.3 PUBLIC HEALTH EVALUATIONS

While there exists a range of benchmarks within clinical medicine (Jin et al., 2020; Pal et al., 2022; Arora et al., 2025), the public health domain, which focuses on areas such as prevention, environmental hazards, community interventions, and managing infectious disease outbreaks, does not have an existing LLM benchmark. In public health, the closest evaluations to those in this paper were performed by Davies et al. (2024), which assessed ChatGPT 3.5's open-ended responses to the UK Faculty of Public Health's Diplomate exam (DFPH) Paper 1 questions. ChatGPT was evaluated on 119 questions double marked by DFPH examiners. Ayers et al. (2023) conduct similar evaluations using 23 questions across 4 public health domains. Finally, work by Harris et al. (2024) evaluated LLMs on classification and extraction tasks using public health guidance free text.

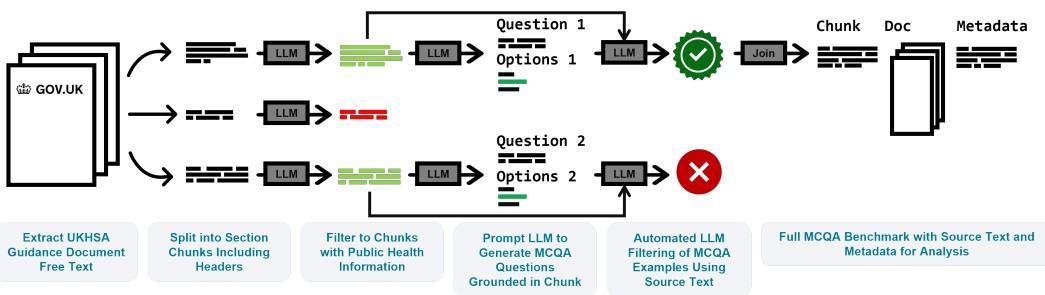

Figure 3: Overview of MCQA generation pipeline.

## 3 METHODS

To generate our MCQA benchmark we develop an automated pipeline (Figure 3) to extract free text from documents, chunk it into sections, generate MCQA samples, and filter to a high quality subset. We focus on an automated approach for three reasons: (1) generating thousands of MCQA samples manually is highly time consuming, (2) public health guidance is frequently revised and so any approach needs to be amenable to regular updates, and (3) it allows us to easily extend our benchmarking to additional knowledge bases in the future.

### 3.1 UK PUBLIC HEALTH INFORMATION

Public health covers a very broad range of topics from biosecurity to tackling health inequalities. To assess LLMs' knowledge of UK guidance across these areas we collect a large corpus of 1,150 current UK Government guidance documents from the UK Government website (gov.uk) in HTML and PDF formats. Using publicly available source documents is necessary for making the benchmark applicable to the real-world and so we assume parts of the document corpus are likely to be included in LLM pre-training datasets. However, by synthetically generating all questions and answer options we can be confident the benchmark was not in model training datasets.

### 3.2 PRE-PROCESSING AND CHUNKING

HTML documents are pre-processed and converted into markdown format. PDF document extraction is more challenging. Therefore, we use a two stage pipeline to achieve the requisite performance on PDF documents. We first extract the raw text from the PDFs using existing tools. We then use OpenAI's GPT-4o-mini vision LLM via the API, prompting the model to extract the text from the image (including markdown headers). For each page individually we pass: an image of the PDF page, the raw markdown text extracted using existing tools for that page at the first step, and the header hierarchy. We then split the documents into 20,488 smaller section chunks based on the markdown headers of each document, and include the hierarchy of higher level headers into every chunk to ensure relevant wider context and document structure is available.

### 3.3 QUESTION GENERATION

Guidance documents often contain significant background information and operational details that do not directly relate to UK public health recommendations. Therefore, in order to generate relevant questions we first use an LLM to classify each chunk into whether it contains a public health recommendation and filter out any chunks that do not. We also filter out chunks exceeding ~2000 words. This reduces the total number of chunks to 7,946, which form the source material for our MCQA generation.

We then use an LLM (Llama-3.3-70bn-Instruct) to generate two multiple choice questions per chunk in the standard MCQA format (Figure 2), with: a single question, single correct answer option, and six incorrect distractor options per question. We use a one-shot with Chain of Thought (CoT) prompt instructing the LLM to output its final answer as a JSON (see Appendix A.3). To ensure the LLM has the required context it is also provided with the text chunks that appear either side of the target chunk in the document (whether or not these contain recommendations). We run this pipeline across all of the filtered text chunks generating 15,666 correctly formatted candidate MCQA samples.

### 3.4 AUTOMATED MCQA ERROR DETECTION AND SAMPLING

To check consistency with the source text and improve the quality of our question set, we use LLMs to filter potentially invalid questions. For full details of the approach and evaluation see Appendix A.1. We select the Llama-3-70bn-Instruct model for this error detection step and filter the 15,666 candidate MCQA questions down to 14,440 that were not flagged as containing potential errors. Finally, we remove questions relating to documents that contain guidance that has been withdrawn, and balance our dataset between HTML and PDF documents to ensure a more even representation of topics (as PDF documents were often longer). We retain the remaining approximately 4,000 unused questions from PDF source documents as a potential internal hold-out set.

## 3.5  FINAL MCQA BENCHMARK DATASET

Overall, the final public benchmark (PubHealthBench) consists of 8,090 MCQA questions covering public, clinical, and professional guidance across 10 public health topic areas, and 352 guidance areas, sourced from 687 documents containing UK Government public health information. Figure 4 provides a breakdown of the benchmark by topic area and the intended audience for the guidance.

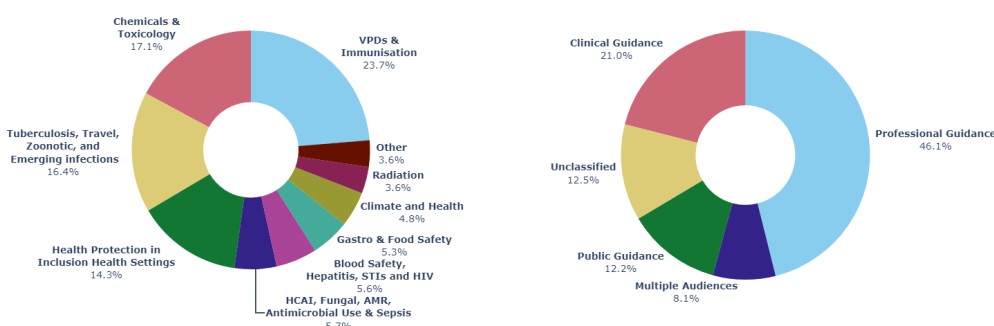

Figure 4: PubHealthBench questions by guidance topic area and guidance audience.

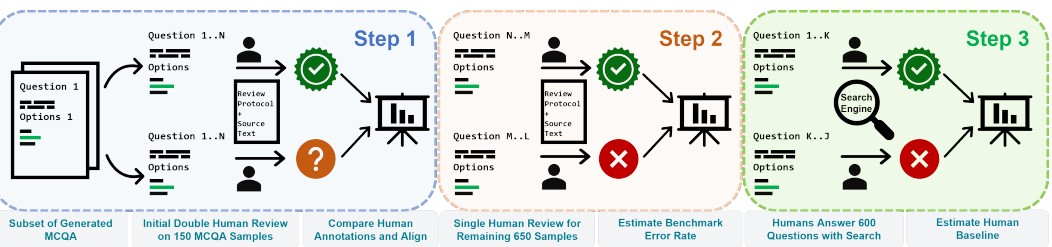

Figure 5: Overview of benchmark review steps.

## 3.6  HUMAN EXPERT QUALITY ASSURANCE

To quality assure the benchmark and estimate the underlying rate of invalid questions, we manually review a random sample of 800 questions (c.10% of the benchmark). Based on this manual review we estimate the rate of ambiguous or invalid questions in the full benchmark to be approximately 5.5% (4.1%-7.3%, 95% Wilson score CI) in the final dataset. See Appendix A.4 for annotation details.

However, the majority of questions identified as invalid related to situations where one of the distractor options could be considered an equally good answer to the specified correct answer. Therefore, whilst these questions were deemed ambiguous, the correct answer should remain one of the most likely guesses for LLMs. Accounting for this random guessing over usually two correct answers, we expect the upper bound score on this benchmark to be approximately 97%. This is supported by the observed model performance on questions classified as invalid, as discussed in Section 5.

## 4  MODEL EVALUATION APPROACH

To understand the level of knowledge current open-weight and proprietary LLMs have about UK Government public health guidance, we assess 24 models on PubHealthBench, including: GPT-4.5 (OpenAI, 2025a), Claude-Sonnet-3.7 (Anthropic, 2025), Gemma-3 (Team, 2025), o1 (OpenAI, 2024), Phi-4 (Abdin et al., 2024), OLMo-2 (OLMo et al., 2025), and Llama-3.3 (Grattafiori et al., 2024).

## 4.1 BENCHMARK SUBSETS

We report overall results for three subsets of PubHealthBench:

**PubHealthBench-Full** - the full MCQA benchmark, providing the broadest assessment of LLM capabilities. This enables us to provide results across granular topic areas within public health.

**PubHealthBench-Reviewed** - the random test subset of 760 MCQA questions that have been manually reviewed by human experts, we report results both including and excluding questions classified as ambiguous or invalid so as to make the results comparable to the full benchmark. For the most expensive models we run this subset instead of the full benchmark for cost reasons.

**PubHealthBench-FreeForm** - the same manually reviewed subset as in *PubHealthBench-Reviewed* but only asking the question (without multiple choice options) and allowing for open-ended free form responses, similar to how a chatbot may respond in real world uses cases. We use a grounded LLM judge to assess whether the free text answer is consistent with the source material.

## 4.2 HUMAN BASELINE

In addition to estimating the theoretical upper bound (97%), we also provide baseline human performance for comparability (Figure 5). Human test takers answered the MCQA samples *with* access to search engines but *without* any LLM or AI enabled tools. Test takers were not trained public health specialists, and were encouraged to take no longer than 2 minutes per question. Under these conditions, humans scored 88% on 600 questions, about 9 percentage points below the potential upper bound. We highlight that this result is meant to simulate a member of the public looking for relevant guidance relatively quickly. See Appendix A.4 for further details on the human baseline setup.

## 4.3 EXPERIMENTAL SETUP

We closely follow the prompts and answer extraction used in the MMLU-Pro benchmark (Wang et al., 2024). However, as with the human baseline, in our LLM evaluations we seek to replicate a similar query setup to that which might occur when interacting with a chatbot or within a simple LLM based application. Therefore, we focus on zero-shot prompting (see Appendix A.6 for the prompt templates), and only use CoT when it is the default behavior, as for reasoning models. For comparability across models and to match the highest risk real world deployments, we do not allow any LLMs access to external tools (e.g search) or information repositories.

### 4.3.1 FREE FORM RESPONSE EVALUATION

Utilising the fact all questions are directly grounded in specific parts of the original source text we use an LLM as a Judge setup (Zheng et al., 2023; Gu et al., 2025) for the free form answer evaluation. We prompt a judge LLM (GPT-4o-Mini) with the question, ground truth answer, the LLM response, and six retrieved related chunks. The judge is asked to assess the response and provide a binary classification for whether it is consistent with the source text and ground truth MCQA answer. For full details see Appendix A.7.

## 5 RESULTS

### 5.1 PUBHEALTHBENCH - OVERALL MCQA BENCHMARK RESULTS

For the 21 models run on the PubHealthBench-Full (Table 1) and the PubHealthBench-Reviewed subset (Table 3), we find a very high correlation in overall accuracy between the two sets, with a correlation coefficient of over 0.99, a rank correlation of 0.98, and an average absolute score difference of under 1 percentage point. Therefore, we compare overall results directly for models only run on PubHealthBench-Reviewed for cost reasons. We provide tables with 95% Wilson score confidence intervals in Appendix A.8.

Table 1: PubHealthBench-Full - zero-shot accuracy for test set of 7929 questions, refusals included as incorrect responses, and bold indicates the highest score. *LLM used to generate benchmark.

| Model Name | Blood Safety, Hepatitis, STIs and HIV | Chemicals and Toxicology | Climate and Health | Gastro and Food Safety | HCAI, Fungal, AMR, Antimicrobial Use and Sepsis | Health Protection in Inclusion Health Settings | Other | Radiation | Tuberculosis, Travel, Zoonotic, and Emerging infections | VPDs and Immunisation | Overall |
|---|---|---|---|---|---|---|---|---|---|---|---|
| GPT-4.5 | **90.9** | **91.1** | **97.4** | **90.3** | **94.6** | **94.0** | **91.3** | **89.9** | **91.8** | **93.0** | **92.5** |
| o3-Mini | 87.7 | 88.5 | 94.5 | 88.1 | 92.4 | 90.6 | 87.5 | 87.1 | 87.1 | 88.3 | 88.9 |
| Gemini-2.0-Flash | 84.7 | 86.1 | 95.3 | 86.0 | 88.2 | 90.6 | 87.1 | 88.5 | 86.3 | 87.4 | 87.7 |
| Llama-3.3-70B* | 86.5 | 86.6 | 92.4 | 85.5 | 87.9 | 90.9 | 86.1 | 87.5 | 85.3 | 87.2 | 87.4 |
| Phi-4-14B | 85.4 | 82.7 | 92.1 | 85.5 | 90.4 | 88.7 | 89.5 | 84.7 | 85.1 | 85.4 | 86.1 |
| Gemini-Pro-1.5 | 81.3 | 81.6 | 93.9 | 84.1 | 87.3 | 90.0 | 86.1 | 85.0 | 84.1 | 86.0 | 85.6 |
| Mistral-3.1-24B | 85.4 | 82.7 | 92.1 | 84.8 | 89.7 | 88.6 | 87.8 | 84.7 | 83.0 | 83.5 | 85.1 |
| GPT-4o-Mini | 83.1 | 80.1 | 91.6 | 81.9 | 88.6 | 88.1 | 86.4 | 82.9 | 79.9 | 82.8 | 83.5 |
| Claude-Haiku-3.5 | 82.4 | 80.5 | 92.4 | 80.3 | 86.2 | 87.6 | 86.4 | 86.1 | 81.1 | 81.5 | 83.2 |
| Gemma-3-27B | 84.5 | 79.7 | 91.6 | 80.3 | 83.9 | 87.4 | 84.3 | 80.5 | 80.1 | 82.0 | 82.7 |
| Gemma-2-27B | 84.2 | 79.3 | 91.3 | 82.9 | 83.9 | 86.9 | 84.3 | 80.5 | 80.4 | 81.3 | 82.6 |
| Phi-4-4B | 82.0 | 79.7 | 90.3 | 81.2 | 87.1 | 85.7 | 86.4 | 77.4 | 78.6 | 80.3 | 81.8 |
| Gemma-3-12B | 80.4 | 79.0 | 89.5 | 76.5 | 83.0 | 84.4 | 86.4 | 81.2 | 79.1 | 79.2 | 80.8 |
| Command-R-32B | 79.7 | 77.2 | 89.5 | 76.7 | 83.7 | 84.8 | 83.6 | 77.7 | 79.1 | 80.7 | 80.7 |
| GPT-4o | 79.5 | 77.4 | 89.2 | 79.1 | 80.8 | 85.9 | 87.5 | 79.8 | 79.6 | 76.7 | 80.2 |
| Llama-3.1-8B | 79.2 | 77.2 | 89.2 | 77.9 | 84.2 | 84.8 | 84.3 | 78.0 | 76.2 | 79.3 | 80.0 |
| Olmo-2-32B | 76.7 | 74.7 | 88.9 | 75.5 | 82.8 | 83.4 | 80.8 | 77.4 | 76.4 | 77.1 | 78.4 |
| Command-R-7B | 76.3 | 72.3 | 86.3 | 74.1 | 76.3 | 79.0 | 82.9 | 71.4 | 70.3 | 73.1 | 74.7 |
| Olmo-2-13B | 76.0 | 69.8 | 87.6 | 74.1 | 76.8 | 78.8 | 80.8 | 74.2 | 71.4 | 72.8 | 74.4 |
| Gemma-3-4B | 73.5 | 69.7 | 85.8 | 71.5 | 73.9 | 79.2 | 81.5 | 67.9 | 69.8 | 67.4 | 72.2 |
| Gemma-3-1B | 49.5 | 49.4 | 57.9 | 45.1 | 50.9 | 51.4 | 57.1 | 44.6 | 44.8 | 42.1 | 47.6 |

On MCQA format questions we find the latest proprietary LLMs perform very strongly, with the highest scoring models GPT-4.5, GPT-4.1[1], and o1[1], all achieving over 90% accuracy, above the human baseline and nearing the benchmark's estimated upper bound.

Smaller open-weight models also show a reasonable degree of knowledge of public health guidance with most 5-15bn parameter models scoring above 75%. However, recent "reasoning" models (o1 and o3-Mini) perform similarly to "non-reasoning" models with little additional benefit from the extra test time compute in the MCQA setting. As shown in Table 3, due to the nature of the MCQA issues identified during manual review, we find models still achieve on average 60% accuracy on MCQA questions labeled invalid. This adds additional support to our estimated upper bound for the benchmark of approximately 97%.

Table 2: PubHealthBench-Full zero-shot accuracy by guidance type. *LLM used to generate benchmark.

| Model Name | Clinical Guidance | Multiple Audiences | Professional Guidance | Public Guidance | Unclassified | Overall |
|---|---|---|---|---|---|---|
| GPT-4.5 | **91.5** | **93.7** | **91.9** | **96.1** | **92.1** | **92.5** |
| o3-Mini | 85.9 | 91.7 | 88.7 | 92.8 | 88.9 | 88.9 |
| Gemini-2.0-Flash | 84.8 | 89.7 | 87.6 | 93.1 | 86.3 | 87.7 |
| Llama-3.3-70B* | 84.9 | 89.5 | 87.1 | 91.7 | 87.1 | 87.4 |
| Phi-4-14B | 84.5 | 88.1 | 85.5 | 90.5 | 85.0 | 86.1 |
| Gemini-Pro-1.5 | 82.5 | 90.0 | 84.8 | 90.6 | 85.5 | 85.6 |
| Mistral-3.1-24B | 81.4 | 87.2 | 84.9 | 90.1 | 86.0 | 85.1 |
| GPT-4o-Mini | 80.7 | 85.8 | 83.1 | 87.9 | 83.9 | 83.5 |
| Claude-Haiku-3.5 | 79.5 | 85.3 | 83.0 | 87.4 | 84.9 | 83.2 |
| Gemma-3-27B | 78.7 | 86.0 | 82.3 | 87.8 | 83.9 | 82.7 |
| Gemma-2-27B | 79.4 | 84.4 | 82.1 | 87.5 | 83.4 | 82.6 |
| Phi-4-4B | 78.3 | 84.4 | 81.8 | 85.4 | 82.5 | 81.8 |
| Gemma-3-12B | 76.9 | 83.2 | 80.7 | 87.1 | 80.2 | 80.8 |
| Command-R-32B | 77.7 | 83.2 | 79.5 | 86.9 | 82.6 | 80.7 |
| GPT-4o | 73.7 | 83.3 | 80.9 | 86.2 | 80.7 | 80.2 |
| Llama-3.1-8B | 76.8 | 82.3 | 80.1 | 82.6 | 81.1 | 80.0 |
| Olmo-2-32B | 74.6 | 81.8 | 78.1 | 84.9 | 77.4 | 78.4 |
| Command-R-7B | 70.1 | 73.6 | 75.6 | 79.3 | 75.4 | 74.7 |
| Olmo-2-13B | 69.6 | 77.2 | 74.2 | 81.2 | 74.9 | 74.4 |
| Gemma-3-4B | 64.7 | 74.2 | 73.1 | 77.7 | 74.7 | 72.2 |
| Gemma-3-1B | 40.1 | 45.2 | 50.3 | 48.7 | 50.5 | 47.6 |

Table 3: PubHealthBench-Reviewed zero-shot accuracy by question and response type. *LLM used to generate benchmark, **Headline result.

| Model Name | Exc. Refusals | Inc. Refusals** | Invalid MCQA | Valid MCQA |
|---|---|---|---|---|
| GPT-4.5 | **92.9** | **92.9** | 71.4 | **94.2** |
| GPT-4.1 | 92.2 | 92.2 | **78.6** | 93.0 |
| o1 | 91.8 | 91.8 | 66.7 | 93.3 |
| Gemini-2.0-Flash | 88.5 | 88.4 | 61.9 | 90.0 |
| o3-Mini | 88.3 | 88.3 | 69.0 | 89.4 |
| Claude-Sonnet-3.7 | 92.4 | 87.8 | 59.5 | 89.4 |
| Llama-3.3-70B* | 87.4 | 87.4 | 61.9 | 88.9 |
| Phi-4-14B | 86.8 | 86.8 | 66.7 | 88.0 |
| Gemini-Pro-1.5 | 86.2 | 86.2 | 59.5 | 87.7 |
| Mistral-3.1-24B | 84.7 | 84.7 | 61.9 | 86.1 |
| GPT-4o-Mini | 83.9 | 83.9 | 52.4 | 85.8 |
| Claude-Haiku-3.5 | 83.3 | 83.3 | 57.1 | 84.8 |
| Gemma-3-27B | 82.9 | 82.9 | 59.5 | 84.3 |
| Gemma-2-27B | 82.9 | 82.9 | 54.8 | 84.5 |
| Phi-4-4B | 81.7 | 81.7 | 64.3 | 82.7 |
| Llama-3.1-8B | 81.1 | 81.1 | 57.1 | 82.5 |
| Command-R-32B | 80.8 | 80.8 | 59.5 | 82.0 |
| GPT-4o | 91.8 | 80.7 | 52.4 | 82.3 |
| Gemma-3-12B | 80.3 | 80.3 | 61.9 | 81.3 |
| Olmo-2-32B | 78.2 | 78.2 | 54.8 | 79.5 |
| Olmo-2-13B | 75.1 | 75.1 | 57.1 | 76.2 |
| Gemma-3-4B | 73.4 | 73.4 | 54.8 | 74.5 |
| Command-R-7B | 72.9 | 72.9 | 57.1 | 73.8 |
| Gemma-3-1B | 45.9 | 45.9 | 26.2 | 47.1 |

## 5.2 PUBHEALTHBENCH - RESULTS BY TOPIC AREA AND AUDIENCE

Breaking down PubHealthBench-Full results by topic area (Table 1), we find consistently higher performance across models on *Climate and Health* and *Health Protection in Inclusion Health Settings* guidance. While LLMs generally performed worse on *Chemicals and Toxicology* (Figure 6).

---

[1]Results on the PubHealthBench-Reviewed subset.

Looking at results by guidance audience (Table 2), we find LLMs have better knowledge of guidance intended for the general public, and worse knowledge of clinical guidance (Figure 6). On public guidance the highest performing model (GPT-4.5) scores 96%, close to the estimated upper bound.

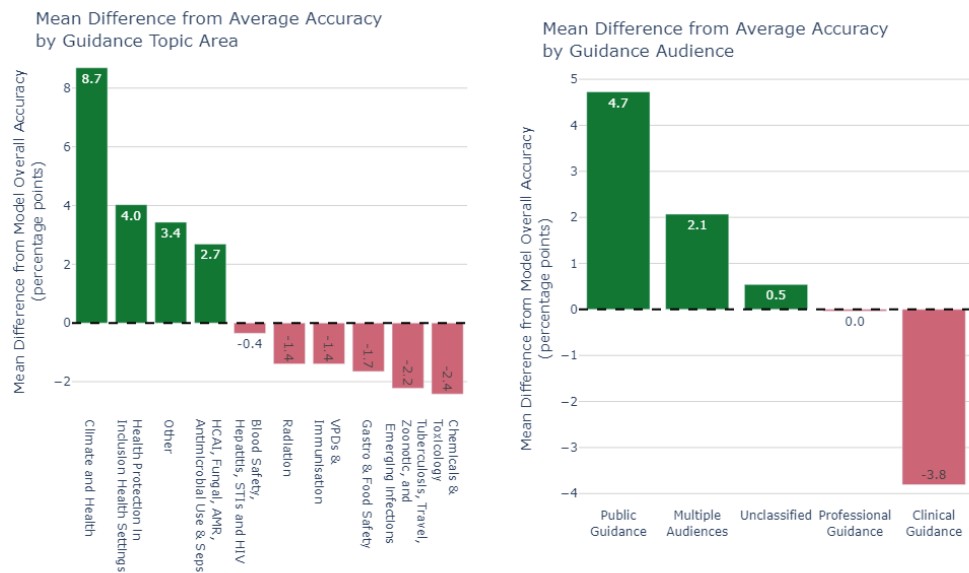

Figure 6: Average deviation from overall model performance on PubHealthBench-Full by guidance topic (left) and guidance audience (right).

### 5.3 PUBHEALTHBENCH-FREEFORM RESULTS

The free form response setup is substantially more challenging for a few reasons: (1) it requires recalling the correct guidance information without any hints from MCQA options, (2) it introduces the possibility of LLMs hallucinating additional information that may be inconsistent with the source text, and (3) the correct answer cannot be inferred via elimination of the other answer options. As a result, all models achieve substantially lower scores in the free form setting by up to 60 percentage points (ppts). The best performing model (o1) scores 74% (Table 4) and also sees the smallest decline from its MCQA performance at -17ppts (Table 5). We provide comparable results using three additional judge models in Appendix A.7, finding a high level of agreement across judges.

Notably there is also significant variation across models with some smaller LLMs (e.g Phi-4-14B) showing over 45ppt declines from MCQA accuracy, while other similarly sized models (e.g Gemma-3-12B) see drops of comparable magnitude to SOTA proprietary LLMs. Also, importantly, as observed in the MCQA setting, LLMs consistently show more accurate knowledge of guidance intended for the general public compared to clinical or professional guidance.

## 6 DISCUSSION

Our results suggest that current SOTA LLMs, both proprietary and open-weight, in general have a very high level of knowledge across UK public health guidance. This is particularly notable given 31% of the MCQA questions were based on guidance documents that were at least partially updated within 2024 (see Appendix A.2), after many of the LLMs' training data cut-off date.

However, we find that performance is significantly degraded across models in the free form response setting. This is in part due to models including extraneous recommendations that do not form part of the source UK public health guidance, but main issues appear to be omitting or contradicting guidance information (see Appendix A.7.2). Qualitative assessment of o1 responses suggests possible problematic outputs are often around the timing of interventions, we provide some examples of these in Appendix A.7.3.

Table 4: PubHealthBench-FreeForm accuracy by guidance audience. *LLM used to generate benchmark, **Judge LLM.

Table 5: Difference in accuracy between MCQA and Free Form settings. *LLM used to generate benchmark, **Judge LLM.

| Model Name | Clinical Guidance | Multiple Audiences | Professional Guidance | Public Guidance | Unclassified | Total |
|---|---|---|---|---|---|---|
| o1 | **71** | **81** | 70 | **86** | **82** | **74** |
| GPT-4.1 | 65 | 71 | **71** | 82 | 69 | 71 |
| o3-Mini | 65 | 78 | 69 | 78 | 74 | 70 |
| GPT-4o | 60 | 71 | 54 | 82 | 63 | 61 |
| GPT-4.5 | 59 | 71 | 54 | 77 | 59 | 59 |
| Claude-Sonnet-3.7 | 57 | 66 | 55 | 76 | 57 | 59 |
| Gemini-2.0-Flash | 56 | 64 | 53 | 77 | 61 | 58 |
| Gemma-3-27B | 50 | 61 | 53 | 73 | 52 | 55 |
| Gemini-Pro-1.5 | 46 | 58 | 51 | 63 | 61 | 53 |
| Gemma-3-12B | 50 | 53 | 48 | 74 | 55 | 52 |
| Claude-Haiku-3.5 | 51 | 61 | 40 | 68 | 47 | 48 |
| GPT-4o-Mini** | 37 | 54 | 40 | 68 | 41 | 43 |
| Llama-3.3-70B* | 35 | 53 | 38 | 60 | 40 | 41 |
| Mistral-3.1-24B | 36 | 39 | 36 | 64 | 40 | 40 |
| Phi-4-14B | 33 | 46 | 37 | 59 | 40 | 39 |
| Olmo-2-32B | 37 | 51 | 37 | 55 | 29 | 39 |
| Gemma-3-4B | 25 | 39 | 35 | 58 | 44 | 37 |
| Command-R-32B | 30 | 41 | 29 | 53 | 42 | 34 |
| Olmo-2-13B | 30 | 32 | 30 | 60 | 34 | 34 |
| Gemma-2-27B | 27 | 36 | 32 | 49 | 34 | 33 |
| Command-R-7B | 20 | 17 | 22 | 46 | 19 | 23 |
| Llama-3.1-8B | 16 | 22 | 15 | 38 | 18 | 19 |
| Phi-4-4B | 16 | 24 | 17 | 29 | 15 | 19 |
| Gemma-3-1B | 14 | 12 | 21 | 26 | 14 | 18 |

| Model Name | PubHealthBench Reviewed | PubHealthBench FreeForm | MCQA - FreeForm Difference |
|---|---|---|---|
| o1 | 91 | 74 | -17 |
| o3-Mini | 88 | 70 | -18 |
| GPT-4o | 80 | 60 | -19 |
| GPT-4.1 | 92 | 70 | -21 |
| Gemma-3-1B | 45 | 18 | -27 |
| Gemma-3-12B | 80 | 52 | -27 |
| Gemma-3-27B | 82 | 54 | -28 |
| Claude-Sonnet-3.7 | 87 | 58 | -29 |
| Gemini-2.0-Flash | 88 | 57 | -30 |
| Gemini-Pro-1.5 | 86 | 52 | -33 |
| GPT-4.5 | 92 | 59 | -33 |
| Claude-Haiku-3.5 | 83 | 48 | -35 |
| Gemma-3-4B | 73 | 36 | -36 |
| Olmo-2-32B | 78 | 38 | -39 |
| GPT-4o-Mini** | 83 | 43 | -40 |
| Olmo-2-13B | 75 | 33 | -41 |
| Mistral-3.1-24B | 84 | 39 | -45 |
| Llama-3.3-70B* | 87 | 40 | -46 |
| Command-R-32B | 80 | 34 | -46 |
| Phi-4-14B | 86 | 39 | -47 |
| Command-R-7B | 72 | 23 | -49 |
| Gemma-2-27B | 82 | 33 | -49 |
| Llama-3.1-8B | 81 | 18 | -62 |
| Phi-4-4B | 81 | 18 | -63 |

Importantly for real-world use cases and deployments we see large disparities between the proprietary and large open-weight models compared with smaller open-weight LLMs (1-15bn parameters). On MCQA questions this gap is generally 10-20ppts but often grows to more than 35ppts in the free form setting. Therefore, there still appear to be significant risks around hallucinations relating to UK public health guidance when using smaller LLMs.

From a public health perspective, it is also an important finding that LLMs consistently performed best on guidance intended for the general public. This audience is likely the highest risk set of users for querying chatbots to retrieve public health information. The fact LLMs are observed to have greater knowledge in this area implies the risks may be lower than the overall results would imply.

## 7 LIMITATIONS

Whilst we have attempted to include a broad range of topics and two LLM response formats (MCQA and free form), a limitation is that this still only represents a few of the ways LLMs could be used to retrieve public health information. Further work investigating different types of public health query is needed, for example, multi-turn interactions, queries including images, queries about topics that are related to public health but not directly incorporated into UK guidance, or allowing tool use.

We use English-language permissively-licensed online UK Government information, which may be incorporated into LLM training data. These results may not extrapolate to LLM performance on other countries' guidance or to other languages. While we hope as the first comprehensive benchmark in this area the approach and LLM performance has broader relevance, the specific results only directly relate to UK information. Further work extending the evaluations to other global guidance sources is needed, particularly for lower resource languages and other global health settings.

## 8 CONCLUSION

In this paper we use an automated pipeline to generate a new benchmark, PubHealthBench, to assess LLM knowledge of current UK Government public health guidance. We evaluate SOTA LLMs across three versions of the benchmark: full, verified, and free form. By testing LLMs across 10 guidance topic areas and 3 intended audiences in the MCQA and free form response setups, we provide an initial assessment of the performance and risks of using current LLMs in this domain. Our results suggest that the latest proprietary LLMs have a high degree of knowledge of UK public health guidance but challenges remain consistently matching the ground truth guidance in the free form setting. We hope this new benchmark will enable further research and evaluation of LLMs for public health, and reduce the risks within real-world deployments.

## 9 ETHICS STATEMENT

All human annotators involved in quality assuring the benchmark were full time employees and paid at least the minimum wage. All data was sourced from the .gov.uk website with permissive licences (Open Government Licence 3.0) and also used in-line with the reuse GOV.UK content terms of service. Large Language Models were used to help generate some diagrams and figures in this paper.

## 10 REPRODUCIBILITY STATEMENT

To ensure reproducibility we provide the full benchmark dataset and example evaluation code as part of the supplementary materials and provide full details of the generation steps in Section 2.3 and Appendix A.6. To enable reproducibility of the free form LLM-as-a-Judge results we also include in the dataset the relevant chunks retrieved from the corpus and provided to the judge.

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

# A  APPENDIX

## A.1  AUTOMATED MCQA ERROR DETECTION

Our approach to automated filtering of candidate questions leverages the fact that synthetic MCQA generation allows us to create an essentially arbitrary number of potential questions but with the risk of material question errors. An advantage of this is that we can utilise an LLM classier with high recall (accurately identifying incorrect questions) even if this comes at the expense of low precision (large numbers of valid questions being rejected) to filter our question set and ensure fewer material errors in the final benchmark.

To assess whether we can use LLMs for this error classification task on our generated MCQA questions we first manually annotated an evaluation dataset of LLM generated samples. We use samples from the earliest version of the pipeline that contained a much higher question error rate to ensure we had enough positive and negative samples. We build on the categorisation approach developed by Gema et al. (2024) to annotate each question with one of 5 categories spanning the common errors that can be found in MCQA questions (the reviewer should allocate the first label that applies):

1. **Valid Question and Options**
   The question contains all the required context to be able to answer the question and it is clear what information the question is seeking.

2. **Ambiguous Question**
   The question cannot be answered standalone for some reason, including:
   
   (a) It is missing context necessary to answer the question (e.g., the disease the question refers to, the population the question refers to, etc.).
   
   (b) It is poorly formed or does not make sense (e.g., asking about a made-up country).
   
   (c) It contains material grammatical or typographical errors.

3. **Ambiguous Options**
   The question is valid but it is not possible to determine if there is a correct option for some reason, including:
   
   (a) The options do not make sense given the question.
   
   (b) The options do not contain enough context to determine whether any are correct.

4. **Incorrect Answer**
   The question is valid and the options are clear but the proposed correct answer (a.) is not a valid correct answer to the question, independent of the other options provided.

5. **Multiple Correct Answers**
   The question is valid, the options are clear, and the proposed correct answer (a.) is true, but there are also other options that would be **equally** valid.

Two human experts then reviewed a random sample of 150 generated questions. We found the level of ambiguity and distractor option differentiation that was permissible for a valid question a challenging task even for expert human annotators. On the 150 questions inter-annotator agreement was 81% with a Cohen's Kappa of 0.39. To create a final validation set of annotations all disagreements were reviewed and resolved by a panel of the two reviewers and a 3rd expert.

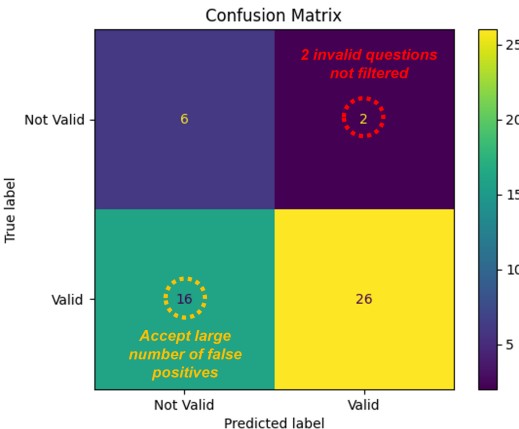

Figure 7: Confusion matrix for LLM MCQA error detection.

For the binary classification task of valid vs invalid, on the test set (of low quality questions) shown in Figure 7 using Llama-3-70bn, this approach would in theory of reduced the benchmark error rate from approximately 16% to approximately 8%. However, we note that the limited test sample means further evaluation of approaches is needed.

In our final pipeline, this stage was less critical as improving both the prompting approach and the LLM used in the generation process (Llama-3 to Llama-3.3) substantially reduced the rate of invalid MCQA samples generated by the pipeline. This is illustrated by the fact only approximately 8% of generated MCQA samples were classified as invalid by the LLM error detection step in the final pipeline, compared with 44% in the validation test set which was sampled from an early version.

## A.2 ADDITIONAL BENCHMARK STATISTICS

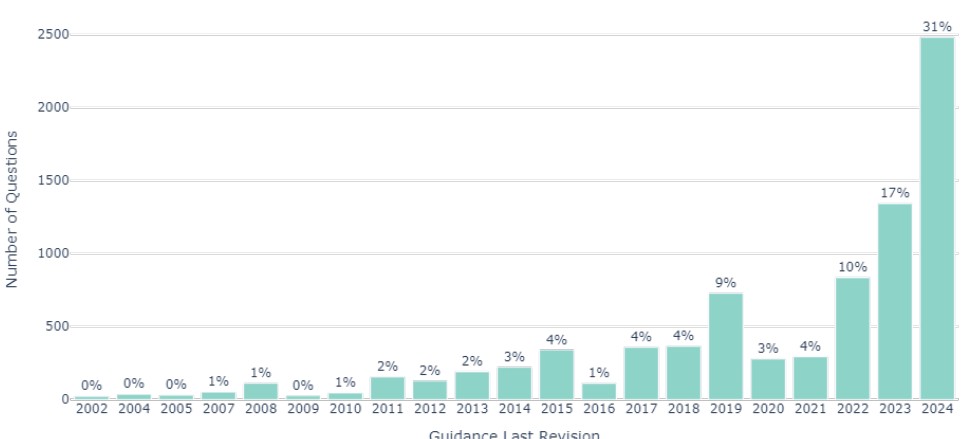

Figure 8: PubHealthBench-Full MCQA by source guidance document last revision date. Note - the last revision to a document may only entail minor changes from the original text.

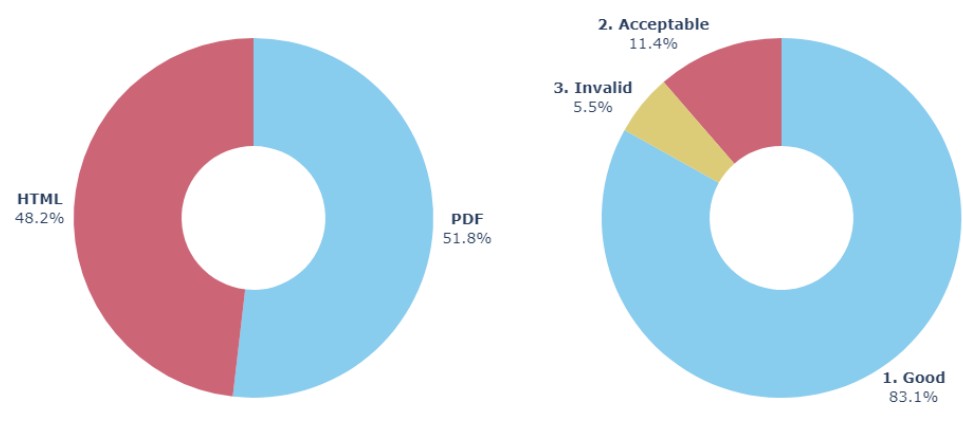

Figure 9: (left) PubHealthBench-Full samples by source text document type, (right) MCQA quality human annotations - "Good" and "Acceptable" categories are treated as valid question samples.

## A.3 MCQA GENERATION PROMPTS

Table 6: MCQA generation system prompt (see Table 7 for user prompt).

---

**System Prompt**

---

You are a top public health expert, creating a multiple choice test to assess individuals' knowledge of UK Public Health Guidance - you should select questions relating to important parts of the information in the guidance text provided that have material public health implications. The questions should not relate to minor details such as phone numbers or other inconsequential information.

To make the questions in the test you are going to be given a piece of current UKHSA guidance, within which there will be a highlighted passage <<START OF SOURCE TEXT THE QUESTION SHOULD RELATE TO >> [PASSAGE] <<END OF SOURCE TEXT THE QUESTION SHOULD RELATE TO >>. You will then do five things:

1. You will provide a list of bullet points of the key parts of highlighted passage in the guidance text provided that could be potentially important public health information and explain why they could be important. These bullet points should entirely come from the highlighted passage in the guidance text provided, you should not include any context or information not contained within the highlighted passage in the guidance text.
2. You will then make a list of bullet points of the relevant pieces of contextual information contained in the highlighted passage of the source text that are necessary to include in future questions - THIS INFORMATION MUST BE INCLUDED IN THE QUESTION ITSELF UNLESS IT IS IRRELEVANT.
3. You will generate a question based on the information in the highlighted passage of the text provided (no other information should be needed to give the correct answer). The public health expert being asked the question will not have the source text and so YOU MUST ENSURE that all relevant context required is included in the question (e.g the names of the dieseases the question relates to, the sub-populations being disucssed, the geographical area the guidance relates to, the time period if specified, etc.).
4. You will then generate 7 multiple choice answers (a. to g.). The correct answer should always be the first answer (a.) you provide. The other answers should be incorrect but very plausible and challenging even for a public health expert to really test whether they actually know the fact. To construct the distractors you should draw on information in both the highlighted passage and the wider guidance provided.
5. Repeat steps 3., and 4., in order to generate a total of 2 questions and answers.

You should provide your final questions and options in the following JSON format:

```
{{
"0": {{"question": [INSERT QUESTION],
"a": [INSERT CORRECT ANSWER],
"b": [INSERT INCORRECT ANSWER],
"c": [INSERT INCORRECT ANSWER],
"d": [INSERT INCORRECT ANSWER],
"e": [INSERT INCORRECT ANSWER],
"f": [INSERT INCORRECT ANSWER],
"g": [INSERT INCORRECT ANSWER]
}},
"1" : {{"question": [INSERT QUESTION],
"a": [INSERT CORRECT ANSWER],
"b": [INSERT INCORRECT ANSWER],
"c": [INSERT INCORRECT ANSWER],
"d": [INSERT INCORRECT ANSWER],
"e": [INSERT INCORRECT ANSWER],
"f": [INSERT INCORRECT ANSWER],
"g": [INSERT INCORRECT ANSWER]
}}
}}
```

---

Table 7: MCQA generation user prompt (see Table 6 for system prompt).

| Prompt Content |
| --- |
| IMPORTANT NOTES YOU MUST FOLLOW:
1. No real phone numbers or urls from the text should be included.
2. In both the question and the answer options you generate you should NOT mention anything relating to the structure of OTHER parts of the source text not included below, for example you should NOT mention things like: other sections of the text (e.g "refer to section 10"), question numbers (e.g "if the patient has answered yes to question 1"), further reading (e.g "see appendix for more information"), etc..
3. Be very careful not to include correct answers in the distractor options b. to g. (or distractors that are potentially correct based on your wider knowledge).
4. You must only generate a question specifically about the passage marked using: << START OF SOURCE TEXT THE QUESTION SHOULD RELATE TO >> [PASSAGE] << END OF SOURCE TEXT THE QUESTION SHOULD RELATE TO >>. You should only use the information outside of this passage for context.
5. Make sure you include all the relevant context in both the questions you generate, these questions will be separated and so should be totally independent.
Here is an example you should use as a template for the structure and style:
==============================
{one shot CoT example}
==============================
Now please follow the instructions above and generate the question and answer options for this piece of UKHSA guidance.
Guidance text:
{guidance text}

Answer (Provide the bulleted list, contextual information, then the final JSON): |

### A.4 Human manual annotation

In this work we perform two sets of human annotation, the first to estimate the rate of invalid or ambiguous questions, and the second to set a human baseline. In both cases, all annotators were full time employees within the organisation with relevant data science or public health experience.

#### A.4.1 Estimating the benchmark error rate

Assessing MCQA sample validity is a challenging annotation task. For a question to be valid, judging the level of context (e.g subpopulation, geography, time period etc.) that is required for the question to be answerable can involve a significant degree of subjectivity. For the options to be valid, the boundary between when a challenging distractor option crosses over into being a potentially equally valid answer to the specified correct answer, rendering the question ambiguous, is also subjective. Therefore, we followed a two round annotation process.

In the first round, pairs of human experts were assigned a total of 150 MCQA samples for double review. Any discrepancies in annotations between reviewers were then assessed by all reviewers and the correct final annotation agreed. This enabled us to identify and rectify inconsistencies in the application of the annotation protocol. In the second round, the remaining 650 MCQA samples were then annotated by single reviewers who participated in the first round.

We provide the Wilson score 95% confidence interval for a binomial proportion. The full instructions provided to reviewers are shown in the box below:

**Protocol for Manual Review of Exam Questions**

This protocol provides structured criteria for assessing the validity of guidance LLM benchmark questions and answer options into three categories: **Good**, **Acceptable**, and **Incorrect**.

**1. Good Questions**
*Criteria:*

1. **Question Valid**: The question is answerable and clearly aligned with the guidance document.

2. **Best Answer Clearly Identifiable**: The correct answer is evidently the 'best' choice compared to the other options and is similar to a hypothetical 'gold standard' answer.

3. **Other Options Incorrect but Not Trivially Wrong**: At least some of the other incorrect options are plausible, not wrong by definition, or so obviously wrong that an uninformed member of the public could say that is not something guidance would ever say.

4. **Informative Value**: The LLM's performance on this question adds meaningful information about the LLM's knowledge of the guidance.

**2. Acceptable Questions**
*Definition:* Questions in this category are valid but have limitations that reduce their overall quality or informativeness. These questions are still useful but might not be as robust as "Good" questions.
*Criteria:*

1. **Question Valid but Some Ambiguity**: The question can be understood and is answerable, but it may be missing some context, contain uncommon acronyms, or may assume a high degree of knowledge.

2. **Gaps in Correct Answer but Still the Best Option**: The correct answer is the best choice from the options provided, but may not be the perfect gold standard answer, may lack some detail or nuance from the guidance, and may be poorly phrased.

3. **Other Options Incorrect but Potentially Trivially Wrong**: All incorrect options are worse options than the correct answer but they may be incorrect trivially, making the correct answer easy to guess.

4. **Some Relevance but Lower Informative Value**: The LLM's performance on this question has some relevance (even if minor) but may test less critical aspects of the guidance, or the question may cover overlapping points with others in the dataset. Very easy questions would also be acceptable.

5. **Not Directly Aligned with the Chunk Provided**: In some cases, the question may relate to text found either side of the intended chunk of guidance in the document. This is acceptable so long as the question still meets the criteria above.

**3. Invalid Questions**
*Definition:* Questions in this category cannot be answered due to material errors and are unsuitable for use in the evaluation.
*Criteria:*

1. **Invalidity**: The question is not answerable due to ambiguity, misinterpretation of guidance, or grammatical errors.

2. **Misleading Answer Options**: The correct answer option provided is either:
   (a) Not a valid answer to the question, or
   (b) Clearly a less accurate answer than one or more of the distractor options.

3. **Errors in Construction**: The question has structural or logical flaws that render it unusable.

### A.4.2 HUMAN BASELINE

To set the human baseline 600 questions were randomly sampled from the manually reviewed subset of 800 questions. To make the human baseline comparable to the full benchmark scores these samples *included* MCQA examples that human annotators had classified as invalid.

The 600 questions were then randomly allocated among 5 human test takers who had not manually reviewed the questions previously. Results were then aggregated to provide the human baseline. Test takers were provided with the following instructions:

---

**Human Test Taking Instructions**

These multiple choice questions are designed to assess LLM knowledge of UKHSA public health guidance. Understanding the performance of humans on these tests is useful for two reasons:

1. To provide a baseline for the overall difficulty of the questions.
2. To understand whether a human asking a chatbot (an LLM) a public health related question is higher or lower risk than a human using previous tools (e.g., Google) to try to find the same information themselves.

To answer the questions, please follow these instructions:

**Question Format**

1. **Standard Multiple Choice** – Every question has one correct answer option in the provided options list.
2. **Select Best Answer** – There may be multiple options that are technically valid information; you should select the answer that is the best reflection of UKHSA guidance.
3. **Don't Know** – For any questions where you aren't confident in the answer, you should provide your best guess.

**Test Conditions**

1. **Timing** – Whilst there is no time constraint on completing the test, we recommend spending no more than 2 minutes on average per question.
2. **Open Book** – Feel free to use your own knowledge and any publicly available online guidance resources to answer the questions.
3. **No LLM Tools** – Please do not use any LLM-based tools or chatbots to search for guidance or question answers.

---

### A.5 EXPERIMENTAL SETUP AND COMPUTE RESOURCES

For all experiments and models where it can be specified we use greedy decoding (temperature 0). For all OpenAI reasoning models we use "reasoning_effort=low" for cost reasons and due to the knowledge based nature of the task.

Proprietary models are accessed via provider APIs, open-weight models are hosted internally on HPC resources using vLLM (Kwon et al., 2023), with up to 2 x A100 80GB used per model. Additional compute (also using 2x A100 80GB) was used for benchmark generation pipeline.

We used a budget of $400 for the proprietary model evaluations, including the LLM judge.

## A.6 PUBHEALTHBENCH PROMPTS

Table 8: MCQA zero-shot evaluation prompt.

**System Prompt**

You are an expert working for a Public Health agency.

**Prompt Content**

The following are multiple choice questions (with answers) about UK Government public health guidance.

Question: This question relates to UK Health Security Agency (UKHSA) guidance that could be found on the gov.uk website as of 08/01/2025.

{question}

Options:
{answer_options}

Provide the letter (A, B, C, D, E, F, or G) of the correct answer. You should state "The answer is (X)", where the X contained in the brackets is the correct letter choice, make sure you include the brackets () around your final answer in your response. DO NOT provide any other information or text in your response.

Answer:

Table 9: Free form evaluation prompt.

**System Prompt**

You are an expert working for a Public Health agency.

**Prompt Content**

The following is a question about UK Government public health guidance.

Question: This question relates to UK Health Security Agency (UKHSA) guidance.

{question}

Please answer the question to the best of your knowledge.

Answer:

## A.7 FREE FORM EVALUATION - LLM AS A JUDGE SETUP

To provide an accurate but cost effective LLM judge to assess unstructured free form LLM responses, we frame it as a binary classification task for whether the free form responses align with the correct answer and source text.

We provide the judge with the guidance source text that was used to generate the question. This ensures the judge always has access to the ground truth information. To allow the judge to correctly evaluate answers that may include additional information that relates to other aspects of UK public health guidance, we provide 5 additional chunks of relevant guidance in the prompt. These 5 ad-

ditional chunks for each question are retrieved from the full corpus using a hybrid retrieval system that ranked relevance by combining the cosine similarities of text embedding vectors (OpenAI's text-3-embedding-large model) and TF-IDF vectors, between each chunk and a given question.

Finally, to guide the judge to the most important information to include, we also provide the MCQA correct answer option. The full LLM judge CoT prompt combining these components is shown in Table 10.

To create a judge evaluation dataset and assess the performance of the LLM judge, we utilise the already generated correct and incorrect MCQA answer options. We insert the known correct or incorrect answer option into a variety of free form chat response templates designed to simulate the structure of LLM free form responses. We assess the ability of the judge to distinguish whether the response is valid using the provided source text.

In this paper we use gpt-4o-mini-2024-07-18 as the main judge with greedy decoding, which achieved over 99% accuracy on the judge evaluation set (10,517 samples). Further details of this LLM judge evaluation approach will be published in upcoming work.

### A.7.1 JUDGE MODEL COMPARISONS

To compare judge models and assess the sensitivity of LLM performance to the judge model used we also re-run the free form evaluation using three additional judge models with range of model sizes. In addition to GPT-4o-Mini, we use Llama-3.3-70bn, MedGemma-27bn, and Phi-4-14bn. Overall we find a high degree of agreement across the four judge models, with a Fleiss's kappa of 0.64, see Figure 10 for full results.

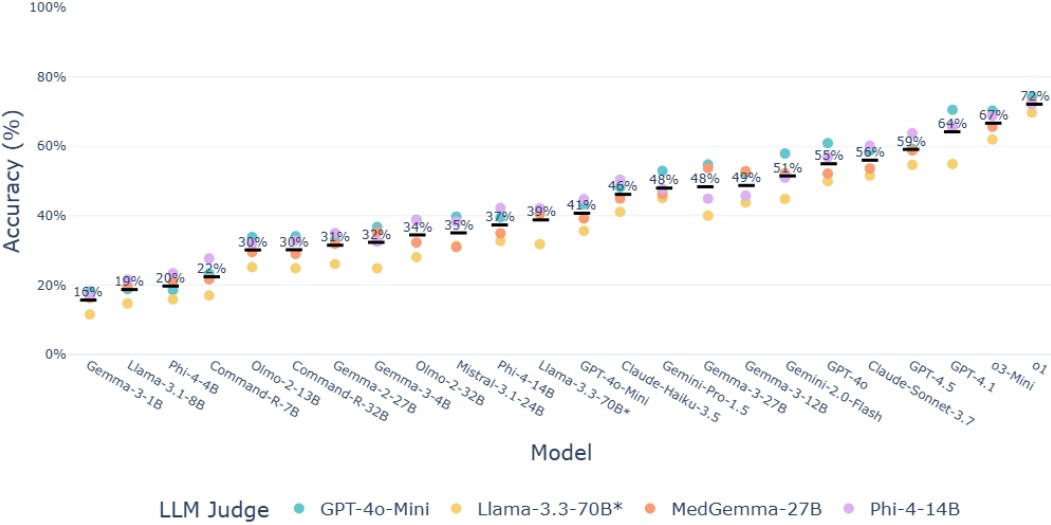

Figure 10: Comparison of PubHealthBench-FreeForm performance using different LLM judges. *LLM used to generate benchmark.

### A.7.2 ERROR ANALYSIS

To understand the categories of errors commonly found in found in free form responses we label the errors in the incorrect LLM responses using Llama-3.3-70bn into the following three types (Figure 11):

1. The answer adds extraneous information not in the official guidance, without contradicting it.

2. The answer omits required points from the official guidance (i.e., is missing information).

3. The answer contradicts or misstates the official guidance (i.e., is an incorrect deviation).

Table 10: Judge prompt.

**System Prompt**

You are an expert in UK public health. You are going to evaluate whether a given answer to a public health guidance question is correct.

**Prompt Content**

You are tasked with evaluating whether a given answer is correct based on the ground truth answer and provided context. Carefully analyse the ground truth answer and context and determine whether the given answer correctly answers the question and aligns with the information given.
—
==========
Question:
==========
{question}
—

==========
Context:
==========
{context}
—

==========
Ground Truth Answer:
==========
{ground truth answer}
—

==========
Given Answer:
==========
{given answer}
—

For the given answer to be correct it must align with the ground truth without omitting any key details **and** any additional detail in the given answer must be seen in the provided context. Determine, with reasoning, whether the given answer is correct based on the ground truth answer and context. Give your response in the following json format:

{{"reasoning": Why is the answer correct/incorrect, "predicted_correct": true or false}}

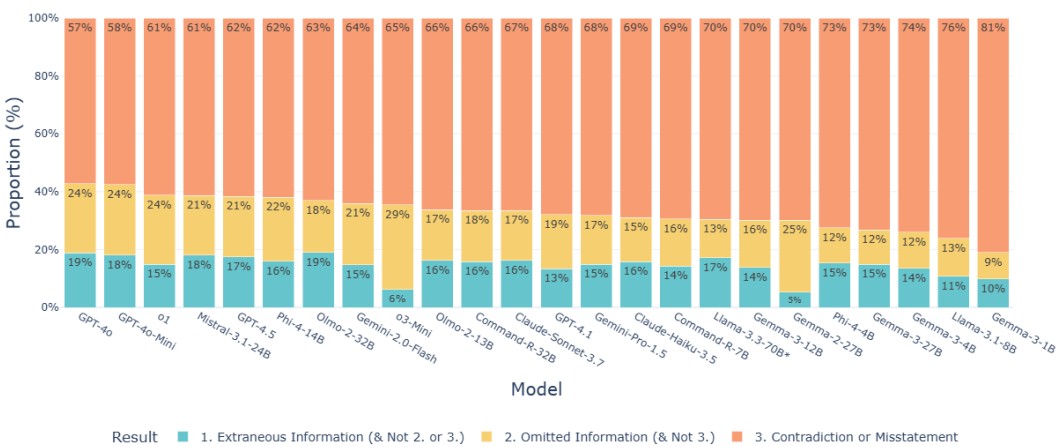

Figure 11: Comparison of PubHealthBench-FreeForm error types classified using Llama-3.3-70bn.

### A.7.3 PROBLEMATIC RESPONSE EXAMPLES

To better understand how even the highest performing LLM responses deviate from the official guidance and the potential types of problem this could cause we also manually reviewed a subset of the errors labeled as category 3. (Section A.7.2) for the top performing model o1.

We identified a number of examples of potentially problematic deviations, which primarily involved suggesting interventions (clinical and public health) too early, too late, or that are not recommended in the guidance. To illustrate the types of mistakes identified and the issues involved we display examples of four different types of problematic response below along with the relevant official guidance passage.

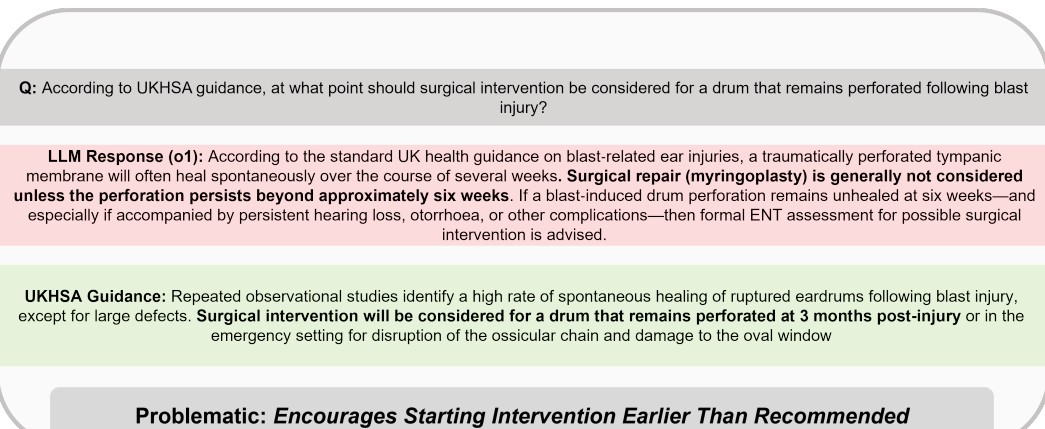

Figure 12: Example of o1 potentially problematic responses to PubHealthBench-FreeForm.

**Q:** According to UKHSA guidance, what is the recommended timing for administering the MenACWY conjugate vaccine to individuals who have already received the MenC conjugate vaccine at the age of 10 years or over?

**LLM Response (o1):** According to the UKHSA's "Green Book" guidance (Chapter 22 on meningococcal vaccination), if someone aged 10 years or older has received a MenC conjugate vaccine, the MenACWY conjugate vaccine should **ordinarily be given at least 12 months after that MenC dose**. However, if the individual is at immediate risk of meningococcal disease, MenACWY can be administered **at a minimum interval of 4 weeks**.

**UKHSA Guidance:** Those who have already received a MenC vaccine over the age of 10 years should still be offered MenACWY conjugate vaccine at the recommended age to ensure protection against the additional capsular groups A, W and Y. The MenACWY conjugate vaccine **can be administered at any interval after MenC vaccine**.

**Problematic:** *Encourages Starting Intervention Later Than Recommended*

Figure 13: Example of o1 potentially problematic responses to PubHealthBench-FreeForm.

**Q:** According to UKHSA guidance, how long should the period of heightened surveillance last after all cases and contacts of a scabies outbreak in a long-term care facility or other closed setting have completed treatment?

**LLM Response (o1):** According to the UK Health Security Agency's guidance, once everyone involved in a scabies outbreak (all cases and their contacts) in a long-term care facility (or other closed setting) has completed their treatment, **a heightened period of surveillance should be maintained for four weeks**. This four-week window helps ensure no new cases emerge and the outbreak has truly been contained.

**UKHSA Guidance:** However, ongoing monitoring and a period of heightened surveillance after all cases and contacts have completed treatment is advised to reduce the risk of outbreaks continuing unchecked.\n\nThis period of heightened surveillance should include regular re-assessment of staff and residents for any new symptoms, and to ensure symptoms are resolving as expected following treatment, and **should last for 12 weeks (that is, 2 mite incubation cycles) after the onset date of symptoms in the last known case**.\n\nA scabies outbreak can be declared over if no new cases are identified within 12 weeks of symptom onset date of the last known case

**Problematic:** *Suggests Ending Intervention Earlier Than Recommended*

Figure 14: Example of o1 potentially problematic responses to PubHealthBench-FreeForm.

**Q:** According to UKHSA guidance, if zanamivir resistance is suspected in a patient being treated for seasonal influenza, what action should be taken regarding their current zanamivir treatment and what additional step is recommended?

**LLM Response (o1):** According to UKHSA guidance, if zanamivir resistance is suspected in a patient being treated for seasonal influenza, **zanamivir treatment should be stopped (i.e. switched to an alternative antiviral),** and a specimen should be sent for antiviral susceptibility testing (via your local diagnostic laboratory to the appropriate reference laboratory).

**UKHSA Guidance:** Several mutations that confer resistance to zanamivir are also associated with resistance or reduced susceptibility to oseltamivir. If zanamivir resistance is believed to be a possibility (for example, as a potential reason for failure to improve**), then continue zanamivir treatment and arrange urgent resistance testing**. Seek advice from local infection specialists. Additional advice is available from regional public health virologists and from the Respiratory Virus Unit at UKHSA.

**Problematic:** *Suggests Additional Intervention*

Figure 15: Example of o1 potentially problematic responses to PubHealthBench-FreeForm.

## A.8 CONFIDENCE INTERVALS

Table 11: PubHealthBench-Full - zero-shot accuracy Wilson score 95% confidence intervals (Bowyer et al., 2025) for test set of 7929 questions, refusals included as incorrect responses, and bold indicates the highest score. *LLM used to generate benchmark.

| Model Name | Blood Safety, Hepatitis, STIs and HIV | Chemicals and Toxicology | Climate and Health | Gastro and Food Safety | HCAI, Fungal, AMR, Antimicrobial Use and Sepsis | Health Protection in Inclusion Health Settings | Other | Radiation | Tuberculosis, Travel, Zoonotic, and Emerging infections | VPDs and Immunisation | Overall |
|---|---|---|---|---|---|---|---|---|---|---|---|
| GPT-4.5 | **(87.8-93.2)** | **(89.5-92.5)** | **(95.2-98.6)** | **(87.1-92.7)** | **(92.2-96.4)** | **(92.4-95.2)** | **(87.5-94.0)** | **(85.9-92.9)** | **(90.1-93.1)** | **(91.8-94.1)** | **(91.9-93.1)** |
| o3-Mini | (84.3-90.4) | (86.7-90.1) | (91.7-96.4) | (84.7-90.9) | (89.6-94.5) | (88.8-92.2) | (83.1-90.8) | (82.7-90.5) | (85.2-88.9) | (86.7-89.6) | (88.2-89.5) |
| Gemini-2.0-Flash | (81.0-87.8) | (84.1-87.8) | (92.6-97.0) | (82.3-89.0) | (84.8-90.8) | (88.8-92.2) | (82.7-90.5) | (84.3-91.7) | (84.3-88.1) | (85.8-88.8) | (86.9-88.4) |
| Llama-3.3-70B* | (83.0-89.4) | (84.7-88.4) | (89.3-94.6) | (81.8-88.6) | (86.4-90.6) | (89.1-92.4) | (81.6-89.6) | (83.1-90.8) | (83.3-87.1) | (85.6-88.6) | (86.7-88.1) |
| Phi-4-14B | (81.8-88.4) | (80.5-84.6) | (89.0-94.4) | (81.8-88.6) | (87.3-92.8) | (86.7-90.4) | (85.5-92.6) | (80.0-88.4) | (83.0-86.9) | (83.7-86.9) | (85.3-86.8) |
| Gemini-Pro-1.5 | (77.4-84.7) | (79.5-83.6) | (91.1-95.9) | (80.3-87.3) | (83.9-90.0) | (88.1-91.6) | (81.6-89.6) | (80.4-88.7) | (82.1-86.0) | (84.8-86.3) | (84.8-86.3) |
| Mistral-3.1-24B | (81.8-88.4) | (80.6-84.6) | (89.0-94.4) | (81.1-87.9) | (86.6-92.2) | (86.6-90.3) | (83.5-91.1) | (80.0-88.4) | (80.8-84.9) | (81.7-85.1) | (84.3-85.9) |
| GPT-4o-Mini | (79.3-86.3) | (77.9-82.2) | (88.4-94.0) | (78.0-85.3) | (85.3-91.2) | (86.1-89.9) | (82.0-89.9) | (77.6-82.0) | (78.8-83.1) | (79.7-83.2) | (82.4-84.0) |
| Claude-Haiku-3.5 | (78.6-85.7) | (78.3-82.5) | (89.3-94.6) | (76.2-83.8) | (82.7-89.1) | (85.6-89.4) | (82.0-89.9) | (81.6-89.6) | (78.8-83.1) | (79.7-83.2) | (82.4-84.0) |
| Gemma-3-27B | (80.8-87.6) | (77.5-81.8) | (88.4-94.0) | (76.2-83.8) | (80.2-87.0) | (85.4-89.2) | (79.7-88.1) | (75.5-84.7) | (77.9-82.2) | (80.2-83.6) | (81.9-83.5) |
| Gemma-2-27B | (80.5-87.4) | (77.1-81.4) | (88.1-93.7) | (79.0-86.2) | (80.2-87.0) | (84.8-88.7) | (79.7-88.1) | (75.5-84.7) | (78.2-82.5) | (79.4-83.0) | (81.7-83.4) |
| Phi-4-4B | (78.1-85.3) | (77.5-81.8) | (86.9-92.9) | (77.2-84.7) | (83.6-89.8) | (83.5-87.6) | (82.0-89.9) | (72.2-81.8) | (76.3-80.7) | (78.4-82.0) | (80.9-82.6) |
| Gemma-3-12B | (76.4-83.8) | (76.7-81.1) | (86.0-92.2) | (72.2-80.3) | (79.3-86.2) | (82.2-86.4) | (82.0-89.9) | (76.3-85.3) | (76.8-81.2) | (77.3-81.0) | (80.0-81.7) |
| Command-R-32B | (75.7-83.2) | (74.9-79.4) | (86.0-92.2) | (72.5-80.5) | (80.0-86.8) | (82.6-86.8) | (78.9-87.5) | (72.5-82.1) | (76.8-81.2) | (78.8-82.4) | (79.8-81.6) |
| GPT-4o | (82.7-86.2) | (75.1-79.6) | (85.7-91.9) | (75.0-82.7) | (76.9-84.2) | (83.8-87.8) | (83.1-90.8) | (74.8-84.0) | (77.3-81.7) | (74.7-78.6) | (79.3-81.1) |
| Llama-3.1-8B | (75.2-82.8) | (74.9-79.4) | (85.7-91.9) | (73.7-81.6) | (80.5-87.2) | (82.6-86.8) | (79.7-88.1) | (72.9-82.4) | (73.8-78.4) | (77.4-81.1) | (79.1-80.9) |
| Olmo-2-32B | (72.5-80.4) | (72.3-76.9) | (85.4-91.7) | (71.2-79.4) | (79.0-86.0) | (81.1-85.4) | (75.9-85.0) | (72.2-81.8) | (74.1-78.7) | (75.1-78.9) | (77.5-79.3) |
| Command-R-7B | (72.1-80.0) | (69.9-74.6) | (82.5-89.4) | (69.7-78.1) | (72.2-80.0) | (76.6-81.3) | (78.1-86.8) | (65.9-76.3) | (67.7-72.7) | (71.1-75.1) | (73.7-75.6) |
| Olmo-2-13B | (71.8-79.8) | (67.3-72.2) | (83.9-90.6) | (69.7-78.1) | (72.7-80.5) | (76.3-81.0) | (75.9-85.0) | (68.9-78.9) | (68.9-73.8) | (70.7-74.7) | (73.5-75.4) |
| Gemma-3-4B | (69.2-77.4) | (67.2-72.1) | (81.9-88.9) | (67.0-75.6) | (69.6-77.7) | (76.7-81.5) | (76.6-85.6) | (62.3-73.1) | (67.3-72.3) | (65.3-69.5) | (71.2-73.2) |
| Gemma-3-1B | (44.9-54.2) | (46.7-52.0) | (52.9-62.8) | (40.4-49.9) | (46.3-55.5) | (48.5-54.3) | (51.4-62.7) | (39.0-50.4) | (42.1-47.5) | (39.9-44.4) | (46.5-48.7) |

Table 12: PubHealthBench-Full zero-shot accuracy Wilson score 95% confidence intervals by guidance type. *LLM used to generate benchmark.

| Model Name | Clinical Guidance | Multiple Audiences | Professional Guidance | Public Guidance | Unclassified | Overall |
|---|---|---|---|---|---|---|
| GPT-4.5 | **(90.1-92.8)** | **(91.5-95.3)** | **(91.0-92.7)** | **(94.7-97.1)** | **(90.2-93.6)** | **(91.9-93.1)** |
| o3-Mini | (84.1-87.4) | (89.3-93.6) | (87.6-89.7) | (91.0-94.2) | (86.8-90.7) | (88.2-89.5) |
| Gemini-2.0-Flash | (83.0-86.4) | (87.1-91.8) | (86.4-88.6) | (91.3-94.5) | (84.1-88.3) | (86.9-88.4) |
| Llama-3.3-70B* | (83.1-86.6) | (86.9-91.6) | (86.0-88.2) | (89.3-93.3) | (84.9-89.1) | (86.7-88.1) |
| Phi-4-14B | (82.7-86.2) | (85.4-90.4) | (83.4-86.6) | (88.5-92.2) | (82.7-87.1) | (85.3-86.8) |
| Gemini-Pro-1.5 | (80.6-84.2) | (87.4-92.1) | (83.6-85.9) | (88.6-92.3) | (83.6-88.0) | (84.8-86.3) |
| Mistral-3.1-24B | (79.5-83.2) | (84.4-89.5) | (83.7-86.1) | (88.0-91.8) | (83.7-88.0) | (84.3-85.9) |
| GPT-4o-Mini | (78.7-82.5) | (82.9-88.3) | (81.8-84.3) | (85.7-89.8) | (81.5-86.1) | (82.7-84.3) |
| Claude-Haiku-3.5 | (77.5-81.4) | (82.4-87.9) | (81.7-84.2) | (85.2-89.3) | (82.6-87.0) | (82.4-84.0) |
| Gemma-3-27B | (76.7-80.6) | (83.1-88.4) | (81.0-83.5) | (85.6-89.7) | (81.5-86.1) | (81.9-83.5) |
| Gemma-2-27B | (77.4-81.3) | (81.4-87.0) | (80.9-83.4) | (85.3-89.4) | (81.0-85.6) | (81.7-83.4) |
| Phi-4-4B | (76.2-80.2) | (81.4-87.0) | (80.5-83.0) | (80.0-84.7) | | (80.9-82.6) |
| Gemma-3-12B | (74.9-78.9) | (80.1-85.9) | (79.4-82.0) | (84.8-89.1) | (77.6-82.6) | (80.0-81.7) |
| Command-R-32B | (75.7-79.7) | (80.1-85.9) | (78.2-80.8) | (84.6-88.9) | (80.1-84.8) | (79.8-81.6) |
| GPT-4o | (71.5-75.8) | (80.3-86.0) | (79.6-82.2) | (83.8-88.2) | (78.1-83.0) | (79.3-81.1) |
| Llama-3.1-8B | (74.7-78.8) | (79.1-85.0) | (78.8-81.4) | (80.1-84.9) | (78.6-83.4) | (79.1-80.9) |
| Olmo-2-32B | (72.4-76.6) | (78.6-84.6) | (76.7-79.4) | (82.5-87.0) | (74.7-79.9) | (77.5-79.3) |
| Command-R-7B | (67.8-72.2) | (70.1-76.9) | (74.2-76.9) | (76.7-81.8) | (72.6-78.0) | (73.7-75.6) |
| Olmo-2-13B | (67.3-71.8) | (73.8-80.2) | (72.8-75.6) | (78.6-83.5) | (72.1-77.5) | (73.5-75.4) |
| Gemma-3-4B | (62.3-66.9) | (70.7-77.4) | (71.7-74.6) | (75.0-80.2) | (71.9-77.3) | (71.2-73.2) |
| Gemma-3-1B | (37.8-42.5) | (41.4-49.1) | (48.7-52.0) | (45.5-51.8) | (47.3-53.6) | (46.5-48.7) |

Table 13: PubHealthBench-Reviewed Wilson score 95% confidence intervals zero-shot accuracy by question and response type. *LLM used to generate benchmark, **Headline result.

| Model Name | Exc. Refusals | Inc. Refusals** | Invalid MCQA | Valid MCQA |
|---|---|---|---|---|
| GPT-4.5 | **(90.8-94.5)** | **(90.8-94.5)** | (56.4-82.8) | **(92.2-95.6)** |
| GPT-4.1 | (90.1-93.9) | (90.1-93.9) | **(64.1-88.3)** | (90.9-94.7) |
| o1 | (89.7-93.6) | (89.7-93.6) | (51.6-79.0) | (91.2-94.9) |
| Gemini-2.0-Flash | (86.1-90.6) | (86.0-90.5) | (46.8-75.0) | (87.6-92.0) |
| o3-Mini | (85.8-90.4) | (85.8-90.4) | (54.0-80.9) | (87.0-91.5) |
| Claude-Sonnet-3.7 | (90.2-94.1) | (85.2-89.9) | (44.5-73.0) | (87.0-91.5) |
| Llama-3.3-70B* | (84.8-89.5) | (84.8-89.5) | (46.8-75.0) | (86.3-91.0) |
| Phi-4-14B | (84.3-89.1) | (84.3-89.1) | (51.6-79.0) | (85.4-90.2) |
| Gemini-Pro-1.5 | (83.5-88.5) | (83.5-88.5) | (44.5-73.0) | (85.1-89.9) |
| Mistral-3.1-24B | (82.0-87.1) | (82.0-87.1) | (46.8-75.0) | (83.3-88.4) |
| GPT-4o-Mini | (81.2-86.4) | (81.2-86.4) | (37.7-66.6) | (83.0-88.2) |
| Claude-Haiku-3.5 | (80.5-85.8) | (80.5-85.8) | (42.2-70.9) | (82.0-87.3) |
| Gemma-3-27B | (80.1-85.4) | (80.1-85.4) | (44.5-73.0) | (81.4-86.7) |
| Gemma-2-27B | (80.1-85.4) | (80.1-85.4) | (39.9-68.8) | (81.7-87.0) |
| Phi-4-4B | (78.8-84.3) | (78.8-84.3) | (49.2-77.0) | (79.8-85.3) |
| Llama-3.1-8B | (78.1-83.7) | (78.1-83.7) | (42.2-70.9) | (79.5-85.1) |
| Command-R-32B | (77.8-83.4) | (77.8-83.4) | (44.5-73.0) | (79.1-84.7) |
| GPT-4o | (89.4-93.6) | (77.7-83.3) | (37.7-66.6) | (79.4-84.9) |
| Gemma-3-12B | (77.3-82.9) | (77.3-82.9) | (46.8-75.0) | (78.3-84.0) |
| Olmo-2-32B | (75.1-80.9) | (75.1-80.9) | (39.9-68.8) | (76.4-82.3) |
| Olmo-2-13B | (71.9-78.1) | (71.9-78.1) | (42.2-70.9) | (72.9-79.2) |
| Gemma-3-4B | (70.2-76.4) | (70.2-76.4) | (39.9-68.8) | (71.2-77.6) |
| Command-R-7B | (69.6-75.9) | (69.6-75.9) | (42.2-70.9) | (70.5-76.9) |
| Gemma-3-1B | (42.4-49.5) | (42.4-49.5) | (15.3-41.1) | (43.4-50.7) |

Table 14: PubHealthBench-FreeForm model accuracy Wilson score 95% confidence intervals by guidance audience. *LLM used to generate benchmark, **Judge LLM.

| Model Name | Clinical Guidance | Multiple Audiences | Professional Guidance | Public Guidance | Unclassified | Total |
|---|---|---|---|---|---|---|
| o1 | **(64.3-77.6)** | **(69.6-89.3)** | (64.9-74.4) | **(76.5-91.9)** | **(72.7-88.3)** | **(71.0-77.2)** |
| GPT-4.1 | (57.8-71.8) | (58.6-81.2) | **(66.1-75.5)** | (72.1-89.0) | (58.8-77.3) | (67.2-73.7) |
| o3-Mini | (57.8-71.8) | (65.9-86.6) | (63.7-73.3) | (67.8-85.9) | (64.5-82.0) | (66.9-73.4) |
| GPT-4o | (52.6-67.0) | (58.6-81.2) | (49.2-59.5) | (72.1-89.0) | (53.3-72.5) | (57.4-64.3) |
| GPT-4.5 | (51.5-65.9) | (58.6-81.2) | (48.3-58.6) | (66.4-84.9) | (49.0-68.6) | (55.7-62.6) |
| Claude-Sonnet-3.7 | (49.7-64.2) | (53.4-76.9) | (49.7-60.0) | (65.1-83.8) | (46.8-66.6) | (55.1-62.1) |
| Gemini-2.0-Flash | (48.6-63.1) | (51.7-75.4) | (47.5-57.8) | (66.4-84.9) | (51.1-70.6) | (54.4-61.4) |
| Gemma-3-27B | (42.4-57.0) | (48.3-72.4) | (47.8-58.1) | (62.3-81.7) | (41.6-61.5) | (51.2-58.2) |
| Gemini-Pro-1.5 | (38.5-53.1) | (44.9-69.4) | (46.1-56.4) | (51.7-72.7) | (51.1-70.6) | (49.3-56.4) |
| Gemma-3-12B | (43.0-57.6) | (40.0-64.7) | (42.7-53.1) | (63.7-82.7) | (44.7-64.6) | (48.8-55.9) |
| Claude-Haiku-3.5 | (44.1-58.7) | (48.3-72.4) | (35.3-45.5) | (57.0-77.3) | (37.5-57.4) | (44.6-51.7) |
| GPT-4o-Mini** | (30.3-44.5) | (41.7-66.3) | (34.8-44.9) | (57.0-77.3) | (31.4-51.0) | (39.8-46.8) |
| Llama-3.3-70B* | (28.2-42.2) | (40.0-64.7) | (32.9-42.9) | (49.2-70.4) | (30.4-49.9) | (37.4-44.3) |
| Mistral-3.1-24B | (29.3-43.3) | (27.6-51.7) | (31.5-41.5) | (53.0-73.9) | (30.4-49.9) | (36.3-43.3) |
| Phi-4-14B | (26.1-39.8) | (33.7-58.3) | (32.6-42.6) | (47.9-69.2) | (30.4-49.9) | (36.1-43.0) |
| Olmo-2-32B | (30.3-44.5) | (38.4-63.2) | (31.8-41.7) | (44.1-65.7) | (20.8-38.9) | (35.4-42.3) |
| Gemma-3-4B | (19.3-32.1) | (27.6-51.7) | (30.7-40.6) | (46.6-68.0) | (34.4-54.2) | (33.4-40.2) |
| Command-R-32B | (23.4-36.9) | (29.1-53.4) | (24.5-33.9) | (41.6-63.3) | (32.4-52.1) | (30.8-37.5) |
| Olmo-2-13B | (24.0-37.5) | (21.7-44.9) | (25.3-34.8) | (49.2-70.4) | (25.5-44.5) | (30.5-37.3) |
| Gemma-2-27B | (20.8-33.9) | (24.6-48.3) | (27.2-36.8) | (37.9-59.6) | (25.5-44.5) | (29.8-36.4) |
| Command-R-7B | (14.7-26.5) | (9.5-28.5) | (18.0-26.6) | (35.5-57.1) | (12.6-28.5) | (20.4-26.4) |
| Llama-3.1-8B | (11.3-22.2) | (13.4-34.1) | (12.1-19.6) | (28.4-49.6) | (11.7-27.3) | (16.2-21.7) |
| Phi-4-4B | (11.3-22.2) | (14.7-36.0) | (13.9-21.8) | (20.5-40.4) | (9.2-23.7) | (15.9-21.5) |
| Gemma-3-1B | (9.4-19.6) | (5.9-22.5) | (16.9-25.4) | (17.3-36.3) | (8.4-22.5) | (15.6-21.1) |

