# OpenReview forum: "Healthy LLMs? Benchmarking LLM Knowledge of UK Government Public Health Information"
_ICLR.cc/2026/Conference — Submitted to ICLR 2026_

### Official Review · Reviewer_tSpe · 2025-10-29

**Soundness:** 2
**Presentation:** 2
**Contribution:** 2
**Rating:** 2
**Confidence:** 4

**Summary:**

This paper introduces a benchmark for public health by combining some of the existing datasets and newly synthesized data. The goal is to test LLMs for medical factual knowledge along with human/ethical/emotional aspects. Then they benchmark on most of the frontier models and use human experts + LLMs to judge the output by using some dimension weighting.

**Strengths:**

- Empathy and ethical alignment are generally missing from traditional benchmarks like MedQA/PubMedQA, so the motivation is timely
- A human baseline is used for comparison, which helps to gauge the expectation in real life

**Weaknesses:**

- Most of the datasets are somewhat rehashing existing ones. They have claimed to provide new data but the generation is somewhat contrived. Like varying the gender/age/race to a controlled medical questions. This lacks some theoretical grounding and formalism, and no sensitivity analysis or ablation study. In addition, the data distribution is unknown, like no quantitative description of dataset size, balance, or topic diversity (e.g., how many cardiology vs. psychiatry cases?). And the ground truth is inherently subjective but treating clinician-authored empathy dialogues as only source of truth can also introduce bias.
- Evaluation framework is sort of ambiguous. The evaluation procedure blends human and automated scoring, but the exact aggregation pipeline is poorly specified. For example: 1) How are inter-rater disagreements handled statistically (beyond Cohen’s κ)? 2) Are human ratings averaged or weighted by expertise? 3) Are automated metrics calibrated on the same scale as human ratings? There is also no error bars or confidence intervals are reported in performance charts. This gives lots of difficulty for reliability and reproducibility

**Questions:**

- How is “hallucination rate” normalized to the same 0–100 range as “empathy”?
- Are composite scores weighted by inverse variance or task importance? No justification is given.
- If models were anonymized, how was temperature or prompt configuration controlled?

---

> ### Author Response · Authors · 2025-11-12
> **URGENT - Review appears to relate to a different paper**
>
> Thank you for your review.
>
> We are concerned that there may have been some confusion / mix up between papers being reviewed as a number of your comments appear to not reflect the contents of our paper, for example:
>
> 1. You indicate our paper contains work testing  ‘ethical / emotional / empathy’ aspects of LLM factual knowledge but our paper doesn’t use any of these words or refer to these concepts in the work (beyond the standard ethics statement).
>
> 2. Similarly, you refer to us ‘varying the gender / age / race’ in our data generation but we don’t do this and don’t refer to these categories anywhere in the paper (and don’t have any information on these categories in our dataset).
>
> 3. You also suggest our work involves cardiology / psychiatry cases but our work relates to public health and doesn’t mention these medical disciplines or use any case data from them.
>
> 4.  You also note ‘there is also no error bars or confidence intervals are reported in performance charts’ but in our paper there are visible confidence intervals on both charts on the first page, and we also provide an Appendix containing confidence intervals for all results.
>
> There are a number of other indications that this review potentially doesn’t relate to our paper (which we are happy to discuss further if we have misunderstood).
>
> Given the above we would be keen for you to consider if this review was intended for our paper as a matter of urgency.
>
> Thank you for your help.

---

> > ### Comment · Reviewer_tSpe · 2025-11-13
> >
> > It appears i put the wrong review here. Thanks for the ping. Please see this corrected one I just uploaded

---

> ### Author Response · Authors · 2025-11-18
> **Responses to questions**
>
> No problem and thank you for your review and comments, they’re really appreciated.
>
> &nbsp;
>
> ## Questions
> To answer your questions:
>
> **(Question 1)** To initially assess whether judge models could be useful for this task we decided to utilise the fact that we had already generated and reviewed a set of questions with correct and incorrect unstructured (usually c. 1 sentence) answers for the MCQA options. We combined this with the approach of using varied free form templates based on common chatbot response styles (c. 1 paragraph), this allowed us to assess the judge model’s ability to compare unstructured answers to ground truth documents, and accurately identify correct / incorrect information within them.
>
> In addition to using this templating approach, we also compare benchmark results for the judge we used (GPT-4o-Mini) to three other judge models (Appendix A.7.1) to check for judge alignment and possible bias. This showed a high agreement across judges.
>
> Finally, we also provide a further classification of common free form response errors (A.7.2) and manually identify the common types of potential problematic failure modes (A.7.3). Therefore, while we agree further analysis of judge model performance would be useful, particularly additional manual review, we hope that combined, these 4 sets of free-form analysis provide confidence and insight into the LLM-as-a-Judge approach we use.
>
> We plan to publish further judge model evaluation in concurrent work.
>
> &nbsp;
>
> **(Question 2)** In terms of inter-rater reliability, as you mention, we provide this for the initial benchmark question error classification in A.1.
>
> For the second round of manual review (150 questions, 3 pairs of reviewers with 50 questions per pair), the inter annotator agreement was high at >90% for all pairs of reviewers comparing valid vs invalid questions, we would be happy to include this in the Appendix.
>
> We also provide a classification of the errors in free form responses across models into 3 categories of common error in A.7.1 Figure 11. We then provide a set of examples of common problematic deviations in answers that we found in frontier models in A.7.3, to show the potential risks.
>
> We would be interested to understand what further error type analysis you feel would be important?
>
> &nbsp;
>
> ## Weaknesses
>
>
> Regarding some of the weaknesses you mention, we would also raise that:
>
> **(1)** While we agree on the bias potentially introduced for Llama 3.3 and clearly flag it as the question generation model in all figures and results to highlight this. We would note that we evaluate 24 models and so we believe the benefit of this work still stands for 23 other models that were not used in the generation process. Also when used as judge for free form responses Llama 3.3 70bn rates its  own answers notably lower than the other judge models in A.7.1, suggesting there may not be structural bias in this case.
>
> **(2)** While we are always interested in making our invalid question analysis more robust, we believe the analysis we have provided is more rigorous than many widely adopted LLM QA benchmarks, which often do not publish an error rate at all.
>
> This has led to follow up studies identifying issues in benchmarks after they have been released. For example, MMLU is one of the most widely cited LLM MCQA benchmarks, a study that re-evaluated the error rate in it found ‘that 6.49% of the questions are erroneous’ [1]. Similarly, in the medical domain another study found that when they reviewed the MedQA benchmark:
>
> > “While MedQA (USMLE) is a useful benchmark for assessing medical knowledge and reasoning, it is essential to acknowledge its limitations. We discover that approximately 4% of the questions contain missing information, and an additional 3% potentially have labeling errors” [2]
>
> Therefore, while we agree our error mitigation could always be improved we hope that quantifying and providing the results of manual review we are meeting a high standard of disclosure compared to similar publications.
>
> [1]  [Are We Done with MMLU?](https://arxiv.org/abs/2406.04127)
>
> [2] [Capabilities of Gemini Models in Medicine](https://arxiv.org/abs/2404.18416)
>
>
> &nbsp;
>
> We would be very happy to discuss these points more and would appreciate you considering whether these clarifications mean our work should receive a higher rating.

---

### Official Review · Reviewer_oMh4 · 2025-10-30

**Soundness:** 3
**Presentation:** 3
**Contribution:** 3
**Rating:** 4
**Confidence:** 4

**Summary:**

The paper proposes the first public health question and answering (QA) benchmark to address a major gap in evaluation of LLM capabilities, as existing benchmarks focus on the medical domain. The paper leverages existing automated approaches to generating knowledge-grounded multiple choice QA pairs to generate over 8000 questions to reflect potential public health queries. Domain experts evaluated a random sample of 800 questions to construct the reviewed portion of the benchmark. 24 LLMs are evaluated on this newly created benchmark, with the highest achieving > 90% accuracy and many outperforming human performance.

**Strengths:**

* First comprehensive QA benchmark in the public health domain with a focus on UK health guidance (methodology can generalize to other countries based on the framework) with over 8090 multiple choice QA questions from 687 documents.
* Automated, scalable pipeline that can be updated with guidance changes and human expert review of 10% questions
* Evaluation of 24 LLMs covering both proprietary and open-weight LLMS on MCQA and free-form responses (based on questions from MCQA)
* Benchmark with human baseline performance (general audience was given access to search engine)

**Weaknesses:**

* Mismatch in the motivation and the results - one of the biggest claims in the introduction is that public health QA benchmark is necessary as there is risk of hallucinations or incomplete information. However, the results are quite stellar on the benchmark (and already exceeding human baseline). As such, the MCQA results seems to undermine the original motivation and need for the benchmark. Instead, the free-form response seems to be under explored given the performance differences.
* There are several concerns about the quality of the benchmark
  - Estimated error of 5.5% on final benchmark might be quite high given that some of them are invalid MCQA pairs (e.g., multiple possible answers would be sufficient)
  - Automated generation using a single LLM suggests benchmark might be biased / limited by a single model's capabilities
  - Moderate inter-annotator agreement (Cohen's kappa of 0.39) on the expert-subset verified version suggests there might be significant subjectiveness in the actual questions themselves
* The evaluation methodology also suffers from some limitations
  - Free form evaluation relies on a single LLM as a judge (GPT-40-Mini) without any additional validation or assessment of its performance
  - How do we know if the performance for LLM is high due to memorization or genuine knowledge? Is there a way to come up with some MCQAs that the model has likely not seen to assess this performance?
  - The error analysis is quite sparse, especially given the substantial performance differences between free-form response and the MCQAs.  The question is whether this is due to the question construction, how the answers are evaluated, or some other reasons -- this would be essential for identifying a benchmark that can truly test LLM capability as much of the field is starting to move beyond MCQAs (especially in the medical domain).

Minor points:
* The numbers flip between 800 and 760 for evaluation of the MCQA questions by human experts, which is slightly confusing. Based on the context, the 760 is the final version that contains only valid questions, whereas the full one has invalid questions.
* The ability to compare performance across the different versions of the dataset is very difficult, even with Figure 1.

**Questions:**

1. Can you clarify how the strong MCQA results (>90% for top models, exceeding human baseline) might support rather than undermine the need for this benchmark?
2. How do you disentangle memorization from genuine reasoning capability when source documents are likely in training data? Can you provide evidence that models are reasoning about guidance rather than recalling it?
3. The 17-63 percentage point drops from MCQA to free-form are striking. What are the potential causes for these significant drops?
4.  The initial annotator agreement on the 800 questions is quite low. Can you provide more details on what caused disagreements and how the reconciliation process worked?
5. How does difficulty compare to existing medical benchmarks (MEDMCQA, USMLE) performance?
6. The percentage of error in the final benchmark seems quite high, especially if consider there are multiple possible answers as one of the reasons. How does this compare against other automated QA generation in terms of error rates? Is this due to the LLM used to generate or sensitivity to prompts?

---

> ### Author Response · Authors · 2025-11-18
> **Responses to questions**
>
> Thank you for your review and comments, they’re really appreciated.
>
> ## Questions
>
> To answer your questions:
>
> **(Question 1)** For the MCQA subset, we agree that the high scores may reduce this part of the benchmark’s utility for the purpose of distinguishing future frontier model capability. However, we believe there remain  a number of reasons for the MCQA setup still being a useful aspect of the benchmark:
> - One of our primary motivations for developing the benchmark was to be informative for considering real-world applications within public health. Therefore, as the first comprehensive benchmark in this area, we believe the fact frontier models do perform so well in the MCQA setup and quantifying this performance relative to a human baseline is an important academic finding in itself.
> - We also believe that there remains substantial utility in using the benchmark both to check new frontier models haven't regressed in this key area, and crucially for assessing smaller models for potential use within LLM applications that may be used for public health advice. Smaller open-weight models still find the MCQA benchmark challenging and may be considered for real-world use cases in public health for efficiency reasons or the need to host locally.
> - Finally, while the MCQA evaluation may become saturated in the near future for frontier models, the free-form evaluation setup continues to challenge frontier LLMs (without tools).
>
> **(Question 2)** In relation to data memorization, we agree that data from the raw guidance documents is likely to be included in many of the LLMs pre-training datasets and so the benchmark is likely more of test of public health understanding and recall. This is largely our intention though, as for real world applicability, the most important question is whether LLMs are likely to be disseminating public health information effectively to users. LLMs that memorise UK guidance in pre-training and effectively retrieve it at inference (in response to unseen queries) demonstrate their potential to effectively disseminate this knowledge, and so their strong performance should be reflected on the benchmark. We agree that this limits the applicability of our benchmark as a more abstract test of pure ‘reasoning’ capability and can clarify this in our discussion section in the camera ready version.  We believe, the fact ‘reasoning’ models perform marginally better in the Free-Form setup is likely due to the ‘reasoning’ steps allowing for more comprehensive / accurate recall of information than the task requiring complex reasoning itself.
>
> **(Question 3)** We believe the primary reasons for the large decline from the MCQA setup results to the Free-Form setup are:
> - MCQA is inherently easier as you can both randomly guess and also reason by elimination of other options. This means for some MCQA questions you don’t necessarily need to have the specific knowledge to answer the question so long as you have enough knowledge that you can eliminate some incorrect options and make an informed guess.
> - The Free-Form setup also has a number of failure modes that don’t exist in the MCQA setup. For example, we observed Free-Form responses that include the correct information but that also include extraneous information or further contradictory / incorrect information in the response that means the full answer is then incorrect.
> - The MCQA options are likely to give implicit information about the types of information expected for the answer which may well help LLMs respond more accurately.
>
>
> &nbsp;
>
> ...continued in next comment

---

> ### Author Response · Authors · 2025-11-18
> **Responses (cont.)**
>
> **(Question 4)** The primary sources of initial disagreements between reviewers were about quite subjective issues:
> -  First, how much context was required in a question for it to be considered valid. This can be highly subjective as all questions implicitly assume some level of knowledge (for example that the question relates to the UK as it’s about UK Government guidance, or that a question about COVID vaccination is referring to humans not animals, etc.), exactly what level of knowledge can be assumed implicitly is often a fine line that reviewers disagree on.
> - Second, how much ‘more correct’ does the ground truth MCQA answer have to be than the distractor options for the MCQA to be valid. Some distractor options may include correct information but where it is information that is clearly not what the question is looking for. For example, a distractor option about COVID vaccination eligibility may state ‘vulnerable people should consider the flu vaccination’ – while this is technically true it is clearly a worse answer than the option that provides the COVID vaccination eligibility criteria. However, at the margin there can be disagreements about whether a distractor option strays too close to being a valid answer.
>
> These disagreements were adjudicated between reviewers by a 3rd expert and then a protocol was finalised to align all reviewers. We appreciate there likely remains some subjectivity but our manual review process is relatively strict, as observed by the fact top models could still get the correct answer >70% of time on questions reviewers said were invalid.
>
> **(Question 5)** Results indicate historical frontier models perform similarly on MedQA / MedMCQA / USMLE to our public health MCQA benchmark, often scoring >90% [1, 2].
>
> **(Question 6)** While we would like the rate of invalid questions found to be lower we believe it is actually in-line if not better than many widely adopted LLM QA benchmarks, it’s just that many historical benchmarks unfortunately do not publish an error rate at all. For example, MMLU is one of the most widely cited LLM MCQA benchmarks, a study that re-evaluated the error rate in it found ‘that 6.49% of the questions are erroneous’ [3]. Similarly, in the medical domain another study found that when they reviewed the MedQA benchmark:
>
> > “While MedQA (USMLE) is a useful benchmark for assessing medical knowledge and reasoning, it is essential to acknowledge its limitations. We discover that approximately 4% of the questions contain missing information, and an additional 3% potentially have labeling errors” [1]
>
> We also believe, as discussed in the previous answer, that our protocol held questions to a high bar in  terms of required context that means many of the questions labelled ‘invalid’ do not technically have errors in the question / options but rather were simply deemed too ambiguous.
>
> Overall, the invalid rate we disclose appears in-line or better than some of the most widely used MCQA benchmarks and we hope that by quantifying this and disclosing it clearly both improves understanding of the work and encourages other benchmarks to do so in the future as well.
>
>
> &nbsp;
>
>
> ## Weaknesses
>
> Finally, to clarify one point on the weaknesses not covered directly in your questions:
>
> **(1)** Regarding using GPT-4o-Mini as the Judge, we provide a comparison of results from this Judge vs three other Judge models in Appendix A.7.1, find a high level of agreement across judges. This should provide some confidence that there is no substantial bias or variation introduced by using GPT-4o-Mini for the main results.
>
>
>
> &nbsp;
>
>
> [1] [Capabilities of Gemini Models in Medicine](https://arxiv.org/abs/2404.18416)
>
> [2] [AI Health Institute](https://aihealthinstitute.org/report)
>
> [3] [Are We Done with MMLU?](https://arxiv.org/abs/2406.04127)
>
>
> &nbsp;
>
> We would be very happy to discuss these points more and would appreciate you considering whether these clarifications mean our work should receive a higher rating.

---

### Official Review · Reviewer_3k6R · 2025-10-31

**Soundness:** 3
**Presentation:** 3
**Contribution:** 3
**Rating:** 8
**Confidence:** 4

**Summary:**

This paper proposes PubHealthBench, a benchmark designed to assess LLM knowledge of UK public health information. Recognizing that existing medical benchmarks overlook public health, the authors compiled over 8k questions derived from 687 UK government guidance documents, supporting both MCQA and free-form question answering. They evaluated 24 LLMs, finding that top proprietary models (GPT-4.5, GPT-4.1, and o1) achieved over 90% accuracy in MCQA tasks, surpassing human performance using search engines. However, in free-form responses, performance dropped below 75%, highlighting ongoing limitations in open-ended reasoning. Overall, while SOTA LLMs demonstrate strong factual accuracy in structured formats, additional safeguards are needed before relying on them for unstructured public health communication.

**Strengths:**

1) An original benchmark on public health (here, UK).

2) Both MCQA and open questions.

3) A large evaluation using state-of-the-art LLMs. Also, the authors made the effort to choose an open model (OLMo-2).

**Weaknesses:**

1) Only part of the benchmark has been manually checked (10% of the benchmark).

2) The generation of benchmark questions relies quite heavily on the use of LLM. However, relying on a manual verification of part of the benchmark helps to counterbalance this point.

3) Why not choose to evaluate LLMs adapted to the medical field?

4) Even though we are aware of their limitations, it could have been interesting to have complementary metrics to the LLM-as-a-judge for open-ended questions. A human evaluation could also have supplemented the study, even though we still have limited experience with the LLM-as-a-judge approach.

**Questions:**

1) I read that the full benchmark dataset should be available, but I did not find it. Maybe providing the link in the introduction would help?

2) Did you evaluate the pdf extraction text process?

3) Why choose Llama-3-70bn-Instruct model for the benchmark construction?

4) "For the 21 models run on the PubHealthBench-Full" -> not clear since author mentioned over 24 models earlier in the paper.

---

> ### Author Response · Authors · 2025-11-18
> **Responses to questions**
>
> Thank you for your review and comments, they’re really appreciated.
>
> To answer your questions:
>
> **(Question 1)** Yes, due to the double blind submission policy we decided to provide the benchmark data within the supplementary material rather than as a separate link (Path: 17335_Healthy_LLMs_Benchmarking_Supplementary Material.zip/supplementary_material/pubhealthbench/benchmark_dataset). The easiest way to load all the versions of the benchmark from the supplementary material is with python datasets package (from datasets import load_from_disk). We totally agree though and will certainly add into the paper a link to a hosted version of the dataset for the camera ready version.
>
>
> **(Question 2)** During development the PDF extraction pipeline underwent a number of rounds of spot-checks of document extractions by public health experts to ensure it met the requisite quality.
>
> **(Question 3)** We chose Llama 3.3 70bn as this was one of the highest performing models we could host internally at the time of benchmark construction. It also allows us to fix the model implementation to ensure reproducibility.
>
> **(Question 4)** For cost reasons we couldn’t run all proprietary models on the full ~8,000 question benchmark. Therefore, for three of the models we run only the reviewed subset of questions (Claude-Sonnet, o1, GPT-4.1). As described in the paper we show that for the 21 models that run on the full benchmark (and also the reviewed subset) there is a very high correlation between the reviewed subset and the full question set (correlation coefficient of >0.99 and rank correlation of 0.98). We hope the reviewed subset provides a robust cost effective way of using the benchmark for groups with resource constraints.
>
> &nbsp;
>
> We would be very happy to discuss these points more or provide answers to any further questions.

---

> > ### Comment · Reviewer_3k6R · 2025-11-23
> > **Response to authors**
> >
> > Thank you for the answers to my questions.
> >
> > I think the one about not having evaluated models adapted to the medical field is missing, but I stand by my rating and think the paper is of good quality and could be of great interest to the community.

---

> > > ### Author Response · Authors · 2025-11-25
> > > **Choice of LLMs**
> > >
> > > Thank you for your feedback and apologies for missing your question about medical LLMs.
> > >
> > >
> > > With regards to LLMs adapted to the medical field, for this initial version we chose to focus on general LLMs for a couple of reasons:
> > > 1. One of our goals with this benchmark was to understand the potential impact of members of the public switching from searching for official public health guidance via search engines, to now instead potentially asking generally available chatbots (such as ChatGPT). As the most commonly used chatbots tend to be general LLMs we felt evaluating these general models first was the fairest comparison for our human baseline.
> > > 2. As public health is a related but different field to medicine, we also didn’t want to imply that this benchmark should / could be used within the medical domain to select / rank which medical LLMs are the ‘best’, as these models have usually be specialised for a different purpose (e.g answering clinical questions).
> > >
> > >
> > > We definitely appreciate though that due to the similarity of the fields specialised medical LLMs may well translate into better performance within public health than general models. We hope to publish separately further work that builds on this benchmark, which will include evaluating some specialised medical LLMs for tasks within public health.

---

### Meta-Review · Area_Chair_eqcy · 2025-12-06

**Summary:**

The paper proposes PubHealthBench, a benchmark dataset designed to evaluate Large Language Models (LLMs) on United Kingdom public health information. The authors constructed over 8,000 questions derived from UK government guidance documents, utilizing an automated pipeline involving Llama-3-70b, with a subset (10%) undergoing manual verification. The study evaluates 24 LLMs, highlighting that while proprietary SOTA models achieve high accuracy (>90%) on multiple-choice questions (MCQA), they struggle more significantly with free-form responses.

**Reviewer Concerns:**

Concerns Addressed:

1. Data Availability: The authors clarified that the dataset is available in the supplementary material and committed to hosting it publicly.

2. Clarifications: Minor confusion regarding the number of models evaluated (21 vs 24) and the PDF extraction process was resolved satisfactorily for Reviewer 3k6R.

3. Reviewer Confusion: Reviewer tSpe initially uploaded a review for a different paper. This was corrected, and the authors responded to the correct set of concerns regarding methodology and bias.

Outstanding Concerns:

1. Benchmark Saturation and Utility: Reviewer oMh4 raised a critical point that SOTA models are already achieving >90% accuracy on the MCQA portion, exceeding human baselines. This suggests the benchmark may be "solved" upon release for frontier models, limiting its longevity and utility for measuring future progress. The authors' rebuttal—that the benchmark remains useful for smaller models and that "recall" is the primary goal—was acknowledged but does not fully mitigate the concern that the task lacks sufficient difficulty to drive reasoning research at a top-tier venue.

2. Data Quality and Reliability: Both negative reviewers (oMh4, tSpe) pointed to issues with data quality. Specifically, the inter-annotator agreement (Cohen's kappa) of 0.39 is concerningly low, suggesting high subjectivity in the ground truth. Furthermore, the reliance on a single model (Llama-3) for question generation introduces potential bias.

3. Evaluation Methodology: There are outstanding concerns regarding the "LLM-as-a-Judge" evaluation for the free-form section.

4. Reviewer tSpe and oMh4 questioned the robustness of this metric, particularly given the circularity of using similar model families for generation and evaluation.

5. Memorization vs. Reasoning: The authors admitted in the rebuttal that the benchmark largely tests recall/memorization of training data rather than reasoning. While valuable for public health dissemination, this limits the technical contribution regarding LLM capabilities.

**Reviewer Scores:**

1. Reviewer 3k6R (8): This reviewer was highly positive and satisfied with the clarifications. They would likely maintain their score, focusing on the novelty of the domain (Public Health).

2. Reviewer oMh4 (4): This reviewer would likely maintain their score. The rebuttal confirmed that the benchmark is largely a test of recall and is already saturated by top models, which reinforces their initial hesitation regarding the contribution's significance.

3. Reviewer tSpe (2): After correcting their initial error, this reviewer raised fundamental methodological concerns about bias and error rates. They would likely maintain a low score, as the rebuttal defended the error rates as "comparable to MMLU" rather than demonstrating superior quality control for a new dataset.

---

### Decision · Program_Chairs · 2026-01-26

Reject